# Talking Turns: Benchmarking Audio Foundation Models on Turn-Taking Dynamics

\* **Siddhant Arora**[1], **Zhiyun Lu**[2], **Chung-Cheng Chiu**[2], **Ruoming Pang**[2], **Shinji Watanabe**[1]
[1] Carnegie Mellon University, USA,[2] Apple
siddhana@andrew.cmu.edu

## Abstract

The recent wave of audio foundation models (FMs) could provide new capabilities for conversational modeling. However, there have been limited efforts to evaluate these audio FMs comprehensively on their ability to have natural and interactive conversations. To engage in meaningful conversation with the end user, we would want the FMs to additionally perform a fluent succession of turns without too much overlapping speech or long stretches of silence. Inspired by this, we ask whether the recently proposed audio FMs can *understand, predict, and perform* turn-taking events? To answer this, we propose a novel evaluation protocol that can assess spoken dialog system's turn-taking capabilities using a supervised model as a judge that has been trained to predict turn-taking events in human-human conversations. Using this protocol, we present the first comprehensive user study that evaluates existing spoken dialogue systems on their ability to perform turn-taking events and reveal many interesting insights, such as they sometimes do not understand when to speak up, can interrupt too aggressively and rarely backchannel. We further evaluate multiple open-source and proprietary audio FMs accessible through APIs on carefully curated test benchmarks from Switchboard to measure their ability to understand and predict turn-taking events and identify significant room for improvement. We will open source our evaluation platform to promote the development of advanced conversational AI systems.

## 1 Introduction

When humans interact with an audio foundation model (FM), it is a two-way communication. Similar to human-human conversations, the model listens and speaks, and more importantly, does both at the same time. Turn management is at the core of real-world conversations. This means that when the user speaks, the system should know when to listen and when to speak (Gravano & Hirschberg, 2011). The system should additionally provide subtle cues, referred to as *backchannels* (Fujie et al., 2005), to indicate that it is "listening" and make relevant follow-up interruptions (Lee & Narayanan, 2010), ensuring that the conversation feels interactive. Furthermore, when the system speaks, it should formulate its output in a way that conveys to the user whether it wants to keep the conversation floor (Ekstedt et al., 2023) and also appropriately address user interruptions (Ma et al., 2024).

Commercial voice assistants (Li et al., 2017) use traditional silence duration-based end-of-turn prediction models for turn management. However, this is insufficient for having *natural* conversations because silence is often not the main cue for humans to switch turns. In fact, research shows that silence within the same speaker's turns (*pauses*) tends to be longer than silence between different speaker's turns (*gaps*) (Ten Bosch et al., 2005). Hence, this continues to remain a challenging task for human-machine interaction. As a result, there has been a lot of interest in automatically predicting turn-taking events. Prior research works (Hara et al., 2018; Fujie et al., 2005) have explored automatically predicting turn change and backchanneling from text, audio, and multimodal input. More recently, audio FMs (Xie & Wu, 2024; Défossez et al., 2024) have been proposed that can automatically take turns while interacting with the end user.

---

\* Work done during an internship at Apple.

However there have been limited efforts to evaluate these FMs on their conversational capabilities. Furthermore, this requires more than simply testing on standard speech recognition (ASR), text-to-speech (TTS), and text-based dialogue benchmarks. To engage in meaningful conversation with the end user, we would want the FMs to additionally perform a fluent succession of turns without too much overlapping speech or long stretches of silence. Finally, the model should be able to engage in dynamic and fluid conversations where it should use cues to confirm listening and encourage the speaker, and it should listen for cues to continue or to be interrupted while speaking. To the best of our knowledge, there has not been any prior effort to empirically evaluate the turn-taking capabilities of audio FMs. As a result, the community still lacks a fine-grained understanding of the relative merits and limitations of different audio FMs on their conversational capabilities. Motivated by this, our work presents the first empirical investigation to evaluate audio FMs on their ability to understand, predict, and perform turn-taking events. Our key contributions are:

**Contribution 1: An evaluation protocol to assess audio FM's capability to *perform* turn-taking events:** Prior work (Nguyen et al., 2023) uses corpus level statistics for automatic evaluation. While these metrics capture how well the distribution of the turn-taking events globally in the generated dialogue matches with ground truth, they fail to evaluate the exact timing when a turn-taking event happens within the local context. We propose evaluating the timing of turn-taking events by training a model on human-human conversations to effectively predict the turn change, backchannel, and interruption. We use this predictor model as the judge since getting human relevance judgments for each turn-taking event is expensive and time-consuming. Using the judge model, we propose *automated* metrics corresponding to each of the core conversational abilities, thus building a comprehensive suite of diverse metrics that can empower a systemic understanding of any audio FM with conversation capability on its ability to manage turns. Finally, we plan to publicly release our evaluation platform so that researchers can easily test their own pre-trained audio FMs.

**Contribution 2: Interesting insights about existing spoken dialogue systems:** We run a user study with different spoken dialogue systems, namely full-duplex E2E spoken dialogue system Moshi and VAD-based cascaded dialogue system and evaluate these systems using both corpus level statistics and our proposed metrics, unveiling many interesting observations: (a) Both dialogue systems sometimes do not speak up even when user wants to yield its turn. (b) Moshi interrupts too aggressively and its interruptions occur at unlikely instances. (c) Both dialogue systems rarely backchannel. (d) Both dialogue systems do not give users enough cues about when they want to keep the conversation floor. (e) Moshi mostly continues speaking even after the user interrupts, whereas a VAD-based dialogue system is more likely to yield its turn when interrupted.

**Contribution 3: Evaluation of audio FMs on understanding and predicting turn-taking events:** We additionally curate a test benchmark using human-human conversation datasets to evaluate audio FMs on their ability to understand and predict turn-taking events. This benchmark facilitates an understanding of the relative merits of different FMs, how this trend varies across different turn-taking events, and quantify the room for improvement.

## 2 RELATED STUDY

**Turn-Taking Prediction**: There have been many prior works on predicting turn change (Gravano & Hirschberg, 2012; Hara et al., 2018; Li et al., 2022), backchannel (Fujie et al., 2005) and interruptions (Lee & Narayanan, 2010; Lee et al., 2008) from audio, text and other multimodal information (Morency et al., 2010; Scherer et al., 2012) (detailed related work discussion in Appendix A.1). The Voice Activity Projection (VAP) models (Ekstedt & Skantze, 2022a; Inoue et al., 2024) are trained in an unsupervised manner on spoken dialogue data to predict future "speech" activity. Other recent works (Wang et al., 2024) use pseudo labels to predict turn-taking and backchanneling locations in spoken dialogue by integrating a neural acoustic model with a large language model (LLM). Recently, audio FMs like Moshi (Défossez et al., 2024) and GPT-4o [1] have been proposed that can perform turn-taking to achieve simultaneous real-time 2-channel conversation. However, these FMs have still not been quantitatively evaluated on their ability to perform turn-taking events.

**Benchmarks for Audio FM**: Benchmarks like Dynamic SUPERB (Huang et al., 2023) and AIR-Bench (Yang et al., 2024) have been valuable for evaluating audio FMs on standard speech process-

---

[1]https://openai.com/index/hello-gpt-4o/

ing tasks, but they do not specifically assess conversational capabilities. SD-Eval (Ao et al., 2024) claims to be an evaluation benchmark for spoken dialogue understanding and generation, evaluating the quality of response generation using automated metrics like BLEU, GPT-4o, or subjective evaluation. However, in this work, we specifically focus on the turn-taking capabilities of audio FMs, particularly their ability to understand, predict, and perform turn-taking events. In addition to previously proposed corpus-level statistics (Nguyen et al., 2023), we introduce our own metrics that evaluate the timing of turn-taking events. These metrics help us gain insights into the limitations of existing systems in engaging in interactive and natural conversations.

## 3    TURN-TAKING ANALYSES BASED ON CONVENTIONAL CORPUS-LEVEL STATISTICS

In the given task setting, we evaluate the recently proposed dialogue systems with turn-taking capabilities using **corpus-level statistics** (Nguyen et al., 2023). The authors organize a user study and compute these statistics from the collected human-AI conversation data.

**Dataset and Audio FM details**: Participants were hired to engage in conversations with the dialogue systems to mirror real-world usage of these systems. As a case study, we evaluate the following spoken dialogue systems: (a) **Moshi**[2], which claims to be a fully E2E duplex spoken dialogue system; (b) **Cascaded** system[3] of open-source models, including Silvero VAD (Team, 2024), Whisper tiny (Radford et al., 2022), SmolLM-135M-Instruct (Allal et al., 2024), and Melo TTS (Zhao et al., 2023c). Our selection of these two systems is driven by the goal of understanding the relative strengths and weaknesses of a fully duplex E2E system compared to a traditional cascaded pipeline. We collect roughly 4 hours of human-AI conversation data for each dialogue system, where each session lasts nearly 5 minutes across 11 different participants. Our analysis confirms that the collected data is sufficient to derive statistically significant insights for all our evaluations (including Sec. 4). The participants primarily included the authors and their research colleagues. To minimize potential variations in results arising from different conversation topics and to facilitate smoother interactions, we provided participants with a predefined list of topics to choose from. More details about our user study in Sec. A.2.

### 3.1    TURN-TAKING EVENTS

Human-human conversations contain a spontaneous succession of turns where overlap and silences occur naturally. Let us assume a two speaker conversation. For simplicity, we divide entire conversation audio into a sequence of $N$ non-overlapping chunks $U = \{U_i \mid i = 1, \ldots, N\}$. Then the voice-activity of the first speaker can be represented by $Y^1 = \{y_i^1 | i = 1, ..N\}$ where $y_i^1 = 1$ if speaker 1 is speaking in chunk $i$. Similarly, the voice-activity of speaker 2 can also be represented as $Y^2$. We show the turn-taking events that occur in spontaneous spoken conversations in Fig. 1.

(1) Inter-Pausal Unit (**IPU**) is a continuous stretch of speech in one speaker's channel, delimited by a silence of more than 200ms from both sides, such that $IPU^k = \{i \in [a,b]|y_i^k = 1\}$, where $a$ and $b$ denote the start and end chunk indexes of $IPU^k$ respectively and k can be 1 or 2.

(2) **Silence** ($SIL$) is part of the conversation where there is no audio from both sides i.e. $SIL = \{i \in [a^{sil}, b^{sil}]|y_i^1 = 0 \text{ and } y_i^2 = 0\}$, where $a^{sil}$ and $b^{sil}$ denote the start and end chunk indexes.

(3) **Pause** is the silence that occurs between the successive IPUs of the same speaker. Successive IPUs by the same speaker separated by a pause are grouped to form a **turn** i.e. $TURN^k = \{\cup_j IPU_j^k |[b_{j-1}, a_j] \subset SIL\}$, where $b_{j-1}$ is the end chunk index of $IPU_{j-1}^k$ and $a_j$ is the start chunk index of $IPU_j^k$.

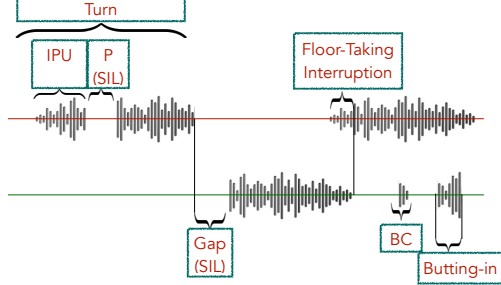

Figure 1: Overview of turn-taking events in human-human conversation

(4) **Gap** is the silence that occurs between IPUs of different speakers, i.e., preceding a **turn change**.

---

[2]https://us.moshi.chat/
[3]https://github.com/huggingface/speech-to-speech

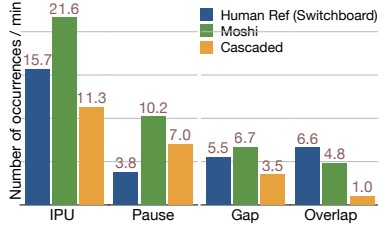 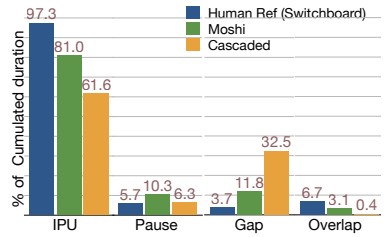

(a) Number of turn-taking events per minute          (b) % of Cumulated durations of turn-taking events

Figure 2: Results of audio foundation models on engaging in conversation with humans based on corpus-level statistics proposed in prior works (Nguyen et al., 2023).

(5) **Overlap** occurs when both speakers talk simultaneously, i.e. $y_i^1 = 1$ and $y_i^2 = 1$.

(6) A **backchannel** is a short utterance such as "um" and "right", which the listener utters without taking the speaker's turn to acknowledge the current speaker. The backchannels typically occur during the speaker's turn. In this work, we denote the backchannel label sequence for the listener $k$ as $BC^k = \{bc_i^k | i = 1, ..N\}$ where $bc_i^k = 1$ if listener $k$ backchannels in chunk $i$.

(7) An **interruption** is defined as overlapping speech between two speakers where both speakers are trying to take the turn. We can also define it in terms of both speaker's IPU such that $IPU^k = \{i \in [a^k, b^k]\}$ and $IPU^{k'} = \{i \in [a^{k'}, b^{k'}]\}$, where $a^k < a^{k'}$ and $a^{k'} < b^k$ such that $k'$ is the interrupting speaker. An interruption can further be classified into (i) **Floor-taking / Successful interruption**: Where the speaker who starts speaking within the IPU of the first speaker takes over the conversation i.e. $b^{k'} > b^k$. (ii) **Butting-in / Unsuccessful Interruption**: Where the speaker who starts speaking within the IPU of the first speaker will be forced to wait for a natural break or pause before speaking, and the first speaker will continue i.e. $b^{k'} < b^k$.

## 3.2 CORPUS-LEVEL STATISTICS

The conversation is passed through a VAD model using pyannotate (Bredin et al., 2020) library. We use the VAD output to get voice activity vectors $Y^{AI}$ and $Y^{Human}$ and analyze the statistics of IPU, Pause, Gap and Overlap events, focusing on the number of events per minute and their cumulated duration as a percentage of the total duration of the conversation. The results are shown in Fig. 2a and Fig. 2b respectively. For reference, we also include statistics derived from human-human conversations in the switchboard dataset (Godfrey et al., 1992), which serve as ground truth. We observe that ① Moshi has a small gap (11.8% of the cumulative duration in Fig. 2b) between speaker turns and some overlapping speech; however, the overlap rate is much lower than that seen in natural human dialogues. ② In contrast, the cascaded system shows higher latency, resulting in a larger gap (32.4% of the cumulative duration) and minimal overlap, making the conversation feel less natural.

While these metrics capture how well the *global* turn-taking events distribution in the generated dialogue matches with ground truth, it cannot evaluate the exact *timing* when a turn-taking event happens. For instance, Fig. 2 shows that Moshi can generate overlapping speech; however, it remains unclear whether this overlap is appropriate and supportive of the user's statements or unexpected and disruptive to the natural flow of the conversation.

## 4 PROPOSED TURN-TAKING ANALYSES WITH TIMING-CENTRIC METRICS

Inspired by the limitations of existing corpus-level statistics, we propose to evaluate the timing of the turn-taking event by training a model that predicts after every 40 ms chunk and is causal, i.e., trained to predict what turn-taking event will happen in the next 40 ms. We then use this predictor model as a proxy of human relevance judgments.

### 4.1 JUDGE TURN-TAKING MODEL

We take inspiration from prior work (Wang et al., 2024) that passes single-channel input speech through the SSL speech foundation model and predicts the listener's behavior as turn-taking decisions that an ideal system should make. However, this approach cannot handle overlapping speech. Hence, we extend their labeling annotation scheme to model both the listener and speaker's behavior and instead use single-channel mixed speech as input. We define the corresponding label sequence

for turn-taking events as $L = \{l_i | i = 1, ..N\}$ where the label for each chunk $l_i$ can be one of the following: (1) **NA**: If no one is talking i.e. $i \in SIL$, (2) **BC** ("Backchannel"): If listener backchannels during the speaker's *turn* i.e. $bc_i^k = 1$ and $i \in TURN^{3-k}$ where k can be either 1 or 2, (3) **I** ("Interruption"): If both speakers are talking to take the turn and none of the speakers are backchanneling i.e. $y_i^1 = 1$ and $y_i^2 = 1$ and $bc_i^1 = 0$ and $bc_i^2 = 0$ (4) **T** ("Turn Change"): If there is a turn change. Special case for interruption: if it follows a floor-taking interruption (see Sec. 3.1), it is also considered a turn change. (5) **C** ("Continuation"): Otherwise.

Our model is *causal* such that it is trained to make future predictions, i.e., what turn-taking event $l_i$ will happen in the *next* chunk by conditioning on only the first $i - 1$ chunks i.e., $U_{1:i-1}$. We further simplify by conditioning only on a prior context window $\hat{U}_i$ of size $W$ (See Eq. 7 in Appendix for more details). The audio in the context window is passed through encoders of the pre-trained speech foundation model, namely Whisper (Radford et al., 2022), to generate acoustic representation $\mathbf{h}_i$. The acoustic representations are then passed through linear layer (Out$(\cdot)$) followed by softmax:

$$\mathbf{f}(\hat{U}_i, \theta) = \text{Softmax}(\text{Out}(\mathbf{h}_i)), \tag{1}$$

where $\mathbf{f}(\hat{U}_i, \theta)[l_i]$ is the likelihood that the turn-taking event $l_i$ will happen at chunk $i$ as predicted by our supervised turn-taking model. More details about the problem formulation and architecture of our model have been provided in Sec. A.3.

**Dataset details:** We train our turn-taking prediction model on Switchboard dataset. Similar to prior work (Wang et al., 2024), we split dataset by conversations into 2000:300:138 for train, validation, and test respectively. We evaluated our model on the in-domain switchboard test set and additionally on 2 out-of-domain (OOD) datasets: the Columbia Games Corpus (Gravano & Hirschberg, 2011) and the

| Turn-Taking Label | Switchboard ($\uparrow$) | Columbia Games ($\uparrow$) | Fisher ($\uparrow$) |
|---|---|---|---|
| **C** (Continuation) | 93.3 | 95.2 | 95.0 |
| **BC** (Backchannel) | 89.4 | 94.0 | 83.3 |
| **T** (Turn change) | 90.8 | 81.6 | 91.6 |
| **I** (Interruption) | 91.3 | 92.6 | 91.8 |
| **NA** (Silence) | 95.1 | 94.0 | 93.5 |
| Overall | 92.0 | 91.5 | 91.0 |

Table 1: Performance (ROC-AUC values) of supervised turn-taking prediction model.

Fisher Corpus (Cieri et al., 2004). We report the model's performance using the ROC-AUC score (Area Under the Receiver Operating Characteristic Curve) similar to Wang et al. The context window of the supervised model $W$ is 30 seconds. More details about the experiment setup and supervised model hyperparameters can be found in Sec. A.4.

**Results and Analysis**: Table 1 shows that our supervised turn-taking prediction model can effectively handle interruptions and performs similar to prior approaches for other turn-taking events, such as those in Wang et al., which reports ROC-AUC scores of 90.29, 81.84, and 91.97 on Switchboard for **C** (continuation), **BC** (backchannel), and **T** (turn change), respectively.[4] Further, the model shows strong OOD generalization, achieving similar performance on the OOD spoken dialogue corpora. This suggests that it can be reliably used to evaluate the precise timing of turn-taking decisions made by AI dialogue systems and serve as a proxy for human relevance judgments.

## 4.2 CORE CONVERSATION CAPABILITIES

To develop metrics that comprehensively evaluate audio FMs' conversational capabilities, we review prior literature (Gravano & Hirschberg, 2011; Skantze, 2021; Raux et al., 2006) and identify the key turn-taking abilities required for a conversational agent to interactively engage with an end user. We design metrics (Sec. 4.4-4.8) corresponding to each capability to ensure complete coverage of the skills required for natural human-AI dialogues. These capabilities are categorized depending on whether they are useful for user input recognition or system output generation, as shown below:

**When user speaks** : (a) *When system should speak up?* : This means that when the user speaks, the system should know when to listen, when to speak, and when to pause, ensuring that the conversation feels natural and not rushed. If the FM speaks too early without much pause, it makes FM sound too eager. On the other hand, if the FM leaves too big of gap, it makes the conversation unnatural. (b) *When system should backchannel?* : The AI system should also provide subtle cues to the user that indicate the assistant is "listening" or processing the user's input. (c) *When system should interrupt?* : The AI system could additionally make relevant follow-up interruptions that encourage further dialogue, making the conversation feel more interactive. However, if it interrupts too aggressively,

---

[4]However, our results are not directly comparable since Wang et al. predicts at word boundaries, whereas we predict at 40ms chunks.

it can come across as rude.

**When system speaks**: (d) *Convey user when it can speak up?* : While generating speech output, it should ensure that it conveys the end user if it wants to yield its turn or keep the conversation floor. (e) *Handle user's interruptions* : When a user interrupts the AI, it should address the interruption or gently steer the conversation back on track.

It is important to note that these conversation capabilities are sensitive to the *timing*, and cannot be evaluated using the corpus-level statistics in Section 3.2. The following section shows that we can create the corresponding metrics by using our judge turn-taking model (Section 4.1).

### 4.3 EVALUATION PROTOCOL

In this work, we design *automated* metrics for each conversation capability using our judge predictor model (Section 4.1). Consistent with prior works on evaluating turn-taking classification models (Ekstedt & Skantze, 2022a), we initially propose metrics that perform a pairwise comparison between 2 turn-taking decisions, for instance, when the user pauses, can the audio FM infer when the user wants to *hold* the conversation floor or when it is ok for assistant to *speak up*. It is important to note that while prior evaluation metrics (Sec. A.1.1) focus on assessing turn-taking model's ability to predict upcoming turn-taking events, our evaluation protocol presents the *first effort to assess audio FMs' ability to perform turn-taking events* in human-AI conversations. We also compute the mean and standard deviation of each metric for each conversation session and show 95% confidence intervals assuming a normal distribution.

**Turn-taking decisions in collected user data**: The sequence of these conversational exchanges is denoted as $U^{\text{dialogue}}$. $Y^{\text{AI}}$ and $Y^{\text{Human}}$ can be computed similar to Sec. 3.2. The input speech for each utterance is passed through an ASR model (Radford et al., 2022) to get the corresponding ASR transcript. We define filler word set as the most frequent isolated one and two-word phrases such as "hmm", "oh", "okay", "i see", etc. We follow Wang et al. and label filler words uttered by the listener in the other speaker's turn as *backchannel*. Using this labeling scheme, we get backchannel sequence $BC^{\text{AI}}$ and $BC^{\text{Human}}$. Using $Y^{\text{AI}}$, $Y^{\text{Human}}$, $BC^{\text{AI}}$, $BC^{\text{Human}}$, we follow our turn-taking label annotation scheme (Sec. 4.1) to compute $L^{\text{dialogue}}$ as the turn-taking decisions that were actually made by the human and AI during the conversation. We can similarly define the turns of the AI dialogue system and the human user as $TURN^{\text{AI}}$ and $TURN^{\text{Human}}$ respectively. Furthermore, we recognize turn-taking events at chunk $i$ as decisions made by the AI system if $i-1 \in TURN^{\text{Human}}$ and as decisions made by Human if $i-1 \in TURN^{\text{AI}}$. This distinction is motivated by prior work (Wang et al., 2024) where the listening system's behavior is considered as turn-taking decisions.

**Judging consistency of metrics**: To quantify the feasibility of using our trained turn-taking model (Section 4.1) as a judge, we take inspiration from prior works (Yang et al., 2024; Zheng et al., 2024) that have experimented with using an LLM as a judge. These studies justify this approach by showing high consistency between LLM predictions and human relevance judgments. Similarly, we evaluate the consistency of our judge model with human judgments by examining instances in a human-human conversation dataset that correspond to each of the proposed metrics. We consider the decisions made by humans during the conversation as a proxy for human judgments. We tune the thresholds defined below for all proposed metrics on an in-domain validation set to maximize the agreement between the judge labels and human judgments (Details in Sec. A.4). Finally, we report the agreement of judge label with human judgments on the in-domain and OOD test sets in Fig. 3 and show that our proposed metrics mostly have high consistency (>60%) with human decisions.

### 4.4 METRIC (A) WHEN USER SPEAKS: WHEN SYSTEM SHOULD SPEAK UP?

To compute this metric, we look at those cases when the user is speaking and then pauses, i.e., $i-1 \in TURN^{\text{Human}}$ and $L_{i-1}^{\text{dialogue}} = \textbf{NA}$. Using $\mathbf{f}(\hat{U}_i^{\text{dialogue}}, \theta)$ (Eq. 1) computed by the judge model, we hypothesize that i) the turn change likelihood $\mathbf{f}(\hat{U}_i^{\text{dialogue}}, \theta)[\textbf{T}]$ should be high for an ideal AI system when it decides to speak up, i.e., $L_i^{\text{dialogue}} = \textbf{T}$ and ii) the continuation likelihood $\mathbf{f}(\hat{U}_i^{\text{dialogue}}, \theta)[\textbf{C}]$ should be high when the system lets user continue i.e. $L_i^{\text{dialogue}} = \textbf{C}$. The judge Label $J_i^1$ is:

$$J_i^1 = \begin{cases} \textbf{T}, & \text{if } \mathbf{f}(\hat{U}_i^{\text{dialogue}}, \theta)[\textbf{T}] - \mathbf{f}(\hat{U}_i^{\text{dialogue}}, \theta)[\textbf{C}] > \text{threshold}_1 \\ \textbf{C}, & \text{otherwise} \end{cases} \quad (2)$$

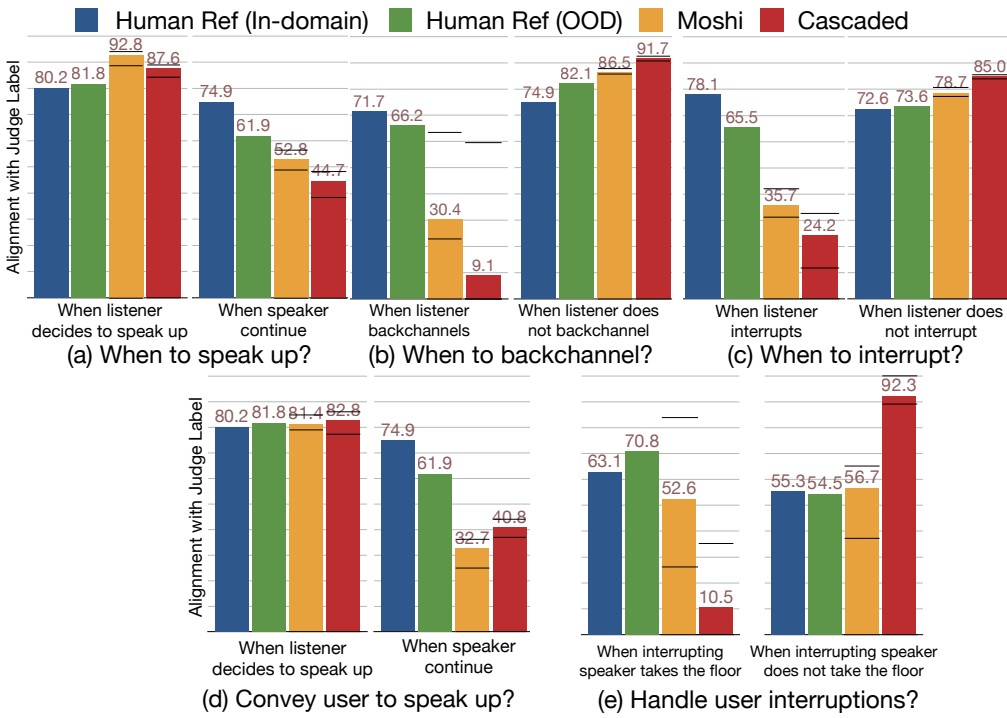

Figure 3: The results show the consistency of the AI dialogue system's turn-taking decisions with judge labels across our proposed metrics. The first 3 graphs correspond to when AI system is *listener* and the remaining 2 graphs correspond to when AI system is *speaker*. Additionally, 95% confidence intervals are provided for AI system with all metrics (also in Appendix Tab. 7). For each graph, the first two bars represent the consistency of our computed judge labels with human relevance judgments obtained from both an in-domain and out-of-domain spoken dialogue corpus.

where threshold$_1$ is a hyperparameter. To compute human judgment, we look at instances where the speaker pauses i.e., $l_{i-1}$ =**NA** and the listener decides whether to speak up i.e., $l_i$ = (**T** or **C**).

**Results and Analysis**: Fig. 3a show that ① Moshi has significantly lower agreement with judge labels compared to humans when it allows users to continue speaking. This finding contrasts with Fig. 2, which indicates that Moshi maintains a gap similar to humans when it initiates speaking, thereby not highlighting any differences in turn-change capabilities. Upon manual inspection, we observed that Moshi occasionally fails to speak up even when the user is ready to yield its turn, consistent with our proposed metric. According to Moshi's architecture (Défossez et al., 2024), there is no explicit boundary to indicate turn changes, i.e., it is listening and generating audio tokens at all time, it simply initiates speaking when it predicts the special token **EPAD** and it stays silent when it predicts the special token **PAD**. Increasing the EPAD logit bias when the user *pauses*—thus encouraging the selection of the EPAD token—could improve Moshi's turn-taking capabilities. ② The cascaded dialogue system performs even worse than Moshi in deciding when to speak up and when to allow the user to continue. It speaks up 37.1% of the time when the user pauses (Table 2), much higher than that seen in human-human conversations, aligning with prior research that silence alone is often not a reliable cue for turn-switching.

### 4.5 METRIC (B) WHEN USER SPEAKS: WHEN SYSTEM SHOULD BACKCHANNEL?

Similarly, we look at when it is the user's turn, i.e., $i - 1 \in TURN^{\text{Human}}$. We hypothesize that i) backchannel likelihood $\mathbf{f}(\hat{U}_i^{\text{dialogue}}, \theta)[\mathbf{BC}]$ should be high when the system decides to backchannel, i.e., $L_i^{\text{dialogue}} = \mathbf{BC}$ and ii) low when the system does not i.e. $L_i^{\text{dialogue}} \neq \mathbf{BC}$. Judge Label $J_i^2$ is:

$$J_i^2 = \begin{cases} = \mathbf{BC}, & \text{if } \mathbf{f}(\hat{U}_i^{\text{dialogue}}, \theta)[\mathbf{BC}] > \text{threshold}_2 \\ \neq \mathbf{BC}, & \text{otherwise} \end{cases} \tag{3}$$

where threshold$_2$ is a hyperparameter. Human judgments are obtained from instances where either the listener begins backchanneling, i.e., $l_{i-1} \neq \mathbf{BC}$ and $l_i = \mathbf{BC}$ or it does not backchannel.

| Dialogue System | AI is Listener | | | AI is Speaker | |
| --- | --- | --- | --- | --- | --- |
| | (a) % Turn Change | (b) % Backchannel | (c) % Interruption | (d) % Turn Change | % Interruption ((e) % Floor-taking) |
| Human Ref | | | | | |
| (in domain) | 15.9 | 0.30 | 0.4 | 15.9 | 0.4 (63.6) |
| (OOD) | 32.9 | 0.09 | 0.4 | 32.9 | 0.4 (60.8) |
| Moshi | 32.7 | 0.01 | 0.5 | 46.5 | 0.2 (17.4) |
| Cascaded | 37.1 | 0.01 | 0.2 | 24.2 | 0.1 (59.4) |

Table 2: Percentage of instances for each proposed metric where the AI makes a specific turn-taking decision. Brackets in the last column show the percentage of interruptions that result in floor-taking.

**Results and Analysis**: Fig. 3b show that both the dialogue systems do ① not backchannel at appropriate points, as indicated by low agreement with judge labels. The confidence intervals are large, which potentially results from a small sample size as ② both systems rarely backchannel (Tab. 2). Moshi's backchanneling capability probably results from its fine-tuning on speech conversations (Défossez et al., 2024) and we recommend increasing the size of this fine-tuning data.

## 4.6 METRIC (C) WHEN USER SPEAKS: WHEN SYSTEM SHOULD INTERRUPT?

Similarly, we hypothesize that i) the interruption likelihood $\mathbf{f}(\hat{U}_i^{\text{dialogue}}, \theta)[\mathbf{I}]$ should be high when the system decides to interrupt, i.e., $L_i^{\text{dialogue}} = \mathbf{I}$ and ii) the continuation likelihood $\mathbf{f}(\hat{U}_i^{\text{dialogue}}, \theta)[\mathbf{C}]$ should be high when the system does not i.e. $L_i^{\text{dialogue}} = \mathbf{C}$. Judge Label $J_i^3$ is:

$$J_i^3 = \begin{cases} \mathbf{I}, & \text{if } \mathbf{f}(\hat{U}_i^{\text{dialogue}}, \theta)[\mathbf{I}] - \mathbf{f}(\hat{U}_i^{\text{dialogue}}, \theta)[\mathbf{C}] > \text{threshold}_3 \\ \mathbf{C}, & \text{otherwise} \end{cases} \tag{4}$$

where threshold$_3$ is a hyperparameter. To compute human judgments, we examine instances where only one speaker is active ($l_{i-1} = \mathbf{C}$) and then observe if both the interrupting and active speakers are talking simultaneously ($l_i = \mathbf{I}$) or if the same speaker continues speaking alone ($l_i = \mathbf{C}$).

**Results and Analysis**: Fig. 3c shows that ① the agreement with judge labels is much lower when interruptions are made by Moshi. While corpus-level statistics (Fig.2) show that both humans and Moshi exhibit overlapping speech, our proposed metric differentiates between them by revealing that interruptions made by humans are expected and collaborative, whereas interruptions made by Moshi are surprising and can be rude. ② The cascaded system rarely interrupts (Table 2) and when it does, they are not well-timed (Fig. 3c), as they typically result from errors in the VAD output.

## 4.7 METRIC (D) WHEN SYSTEM SPEAKS: CAN IT CONVEY WHEN USER CAN SPEAK UP?

To compute this metric, we look at when the system is speaking and then pauses, i.e., $i-1 \in TURN^{\text{AI}}$ and $L_{i-1}^{\text{dialogue}} = \mathbf{NA}$. We hypothesize that the i) turn change likelihood $\mathbf{f}(\hat{U}_i^{\text{dialogue}}, \theta)[\mathbf{T}]$ should be high when the user believes that an ideal AI system can speak up, i.e., $L_i^{\text{dialogue}} = \mathbf{T}$ and ii) the continuation likelihood $\mathbf{f}(\hat{U}_i^{\text{dialogue}}, \theta)[\mathbf{C}]$ should be high when the system continues i.e. $L_i^{\text{dialogue}} = \mathbf{C}$. Judge label ($J_i^1$ in Eq. 2) and human relevance judgments are calculated similar to Metric **(1)** (Sec. 4.4).

**Results and Analysis**: Fig. 3d demonstrates that ① Moshi does not give users enough cues when it wants to keep the conversation floor, leading to the lower agreement with judge labels and the user speaks up more frequently (46.5% in Tab. 2) when Moshi pauses. Similar to Sec. 4.5, fine-tuning on additional speech conversations could improve its turn willingness capability. ② The cascaded dialogue system was slightly better at conveying turn willingness than Moshi.

## 4.8 METRIC (E) WHEN SYSTEM SPEAKS: HANDLE USER INTERRUPTIONS?

Ma et al. has proposed Interactive capability metric to evaluate audio FM's capability to handle user's interruptions, which looks at whether the system yields its turn when the user interrupts. Motivated by this, we look at instances when the user has already made an interruption, i.e., $L_{i-1}^{\text{dialogue}} = \mathbf{I}$. We hypothesize that i) the turn change likelihood $\mathbf{f}(\hat{U}_i^{\text{dialogue}}, \theta)[\mathbf{T}]$ should be high when ideal AI system lets the user take the conversation floor (Sec. 3.1), i.e., $L_i^{\text{dialogue}} = \mathbf{T}$ and ii) the continuation likelihood $\mathbf{f}(\hat{U}_i^{\text{dialogue}}, \theta)[\mathbf{C}]$ should be high when it keeps the floor i.e, $L_i^{\text{dialogue}} = \mathbf{C}$. Judge Label $J_i^4$:

$$J_i^4 = \begin{cases} \mathbf{T}, & \text{if } \mathbf{f}(\hat{U}_i^{\text{dialogue}}, \theta)[\mathbf{T}] - \mathbf{f}(\hat{U}_i^{\text{dialogue}}, \theta)[\mathbf{C}] > \text{threshold}_4 \\ \mathbf{C}, & \text{otherwise} \end{cases} \tag{5}$$

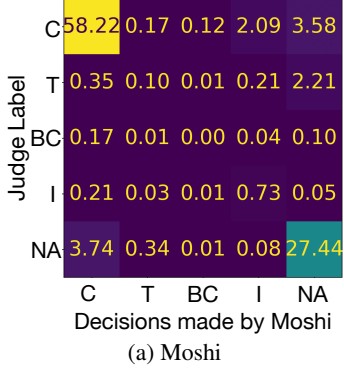
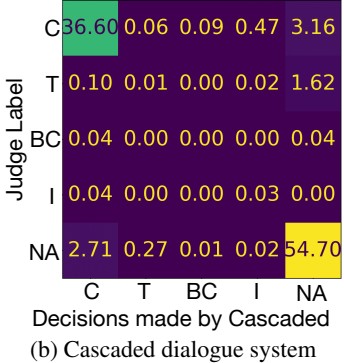

| (a) Moshi | (b) Cascaded dialogue system |
|---|---|

Figure 4: Confusion Matrix showing the performance of the turn-taking decisions $L^{\text{dialogue}}$ made by the AI systems using the supervised turn-taking model as judge (i.e. $L^{\text{gen}}$ as the ground truth). The numbers in the confusion matrix represent percentages.

where $\text{threshold}_4$ is a hyperparameter. To compute human relevance judgments, we identify instances where both the interrupting and active speakers are simultaneously speaking ($l_{i-1} = \mathbf{I}$) and then evaluate the consistency of the judge's label with speakers decision to allow a turn change.

**Results and Analysis**: Compared to other metrics, judge labels have lower consistency with human relevance judgments for this particular metric as shown in Fig. 3e, suggesting that insights derived using this metric may be less reliable. ① Tab. 2 shows that only 17.4% of user interruptions led to a turn change, with Moshi continuing to speak most of the time. Increasing the **PAD** (See Sec. 4.4) logit bias when the user *interrupts* could encourage Moshi to occasionally become silent, which could help it to avoid ignoring user interruptions entirely. ② Moshi's decisions do not differ significantly in agreement with judge labels compared to humans. ③ The cascaded dialogue system yielded its turn more than 50% of the time (Tab. 2) when the user interrupted, even when the user was not trying to take over the conversation but merely supporting the AI's thoughts. However, the number of user interruptions is too few (0.1% as shown in Tab. 2) to draw strong conclusions, resulting in large confidence intervals in Fig. 3e.

## 4.9 SINGLE-LABEL EVALUATION

Additionally, we use our judge model in single answer grading setup (Zheng et al., 2024) where we directly assign a turn-taking event label to each chunk, thus generating the pseudo ground truth labels $L^{\text{gen}}$ by tuning operating points (thresholds, see Sec. A.4) for the predicted likelihood on the validation set.[5] Using the generated ground truth labels $L^{\text{gen}}$, we compute a confusion matrix of $L^{\text{dialogue}}$ for decisions made by an AI system. Fig. 4a and 4b show the confusion matrices for the decisions made by Moshi and the cascaded dialogue system, respectively. ① Moshi has a high rate of false positives for interruptions (2.09% of total decisions), indicating that it interrupts the user too aggressively when it should have allowed the user to continue speaking (i.e., **C**). Decreasing the **EPAD** logit bias (Défossez et al., 2024) when the user is actively speaking could reduce the frequency of these interruptions, making Moshi's behavior less rude. ② Moshi is also not effective at backchanneling, with our judge model predicting few opportunities for it in human-AI interactions. ③ The accuracy of turn changes is low, mainly due to false negatives (2.21% of total decisions), meaning Moshi often fails to speak up when it should. ④ For the cascaded system, there are too few possibilities of either an interruption or backchannel showing that the conversation is not interactive.

## 5 ADDITIONAL EVALUATION ON UNDERSTANDING AND PREDICTING TURN-TAKING EVENTS

While most existing open source and proprietary audio FMs cannot perform turn-taking events, many audio FMs (Chu et al., 2024; 2023) claim multi-turn dialogue capabilities where end-of-turn is manually specified by the user. Hence, we additionally investigate whether training on multi-turn audio dialogues can enable these FMs that *cannot perform turn-taking events* to at least understand the meaning of turn-taking events and recognize the acoustic and semantic cues that precede their occurrence. Sec. A.5 in the Appendix contains details about experiment setup and audio FMs.

---

[5]Due to the significant class imbalance among turn-taking events, using a simple argmax is not suitable.

**How Well Audio Foundation Models Understand Turn-Taking?:** In this task setting, the audio FM is provided with the first $i$ chunks of audio and must predict the turn-taking event that occurred within the given audio. We focus on three

| Model | Turn Change (↑) | Backchannel (↑) | Interruption (↑) |
|---|---|---|---|
| Random Baseline | 50.0 | 50.0 | 50.0 |
| SALMONN | 41.4 | 50.1 | 51.3 |
| Qwen2-Audio-Instruct | 48.8 | 48.5 | 50.3 |
| Qwen-Audio-Chat | 56.7 | 52.7 | 69.5 |
| Whisper+GPT-4o | 66.3 | 49.1 | 52.9 |

Table 3: Accuracy of audio foundation models on test benchmarks evaluating their ability to *understand* turn-taking events

specific turn-taking events: turn change **T**, backchannel **BC**, and interruption **I**. We present the performance of audio FMs in Table 3. Among the open-source audio FMs, Qwen-Audio-Chat achieves the best performance. A cascade of Whisper and GPT-4o generally outperforms the open-source models. However, all models perform close to a random-guess baseline when it comes to understanding backchannels.

**How Well Audio Foundation Models Predict Turn-Taking?:** In the given task setting, the audio FM is provided access to the first $i-1$ chunks of audio, and it has to predict the turn-taking event that will happen next i.e. $l_i$. We focus on predicting four specific

| Model | Turn Change (↑) | BackChannel (↑) | Interruption (↑) | Floor-Taking Interruption (↑) |
|---|---|---|---|---|
| Random Baseline | 50.0 | 50.0 | 50.0 | 50.0 |
| Supervised Topline (Sec. 4.1) | 78.6 | 75.1 | 74.9 | 65.6 |
| SALMONN | 49.3 | 50.0 | 50.0 | 50.4 |
| Qwen2-Audio-Instruct | 46.5 | 49.3 | 51.5 | 54.4 |
| Qwen-AudioChat | 49.9 | 52.1 | 52.3 | 50.8 |
| Whisper+GPT-4o | 62.2 | 48.6 | 49.3 | 50.0 |

Table 4: Accuracy of audio foundation models on test benchmarks evaluating their ability to *predict* future turn-taking events.

turn-taking events: turn change, backchannel, interruption, and whether the interruption is *floor-taking*. To identify areas for improvement, we compare the audio FM's performance against our supervised turn-taking model, using $J_i^1$ (Eq.2), $J_i^2$ (Eq.3), $J_i^3$ (Eq.4) and $J_i^4$ (Eq.5) to predict turn change, backchannel, interruption, and floor-taking interruption, respectively. We present our findings in Table 4. The cascade of Whisper and GPT4-o API is very good at predicting turn change, which is not very surprising since prior works (Ekstedt & Skantze, 2020) have shown that text-based LM can effectively predict turn shifts. However, it cannot predict backchannel, interruptions, and whether the interruption will lead to turn change. All the open-source audio FMs perform close to a random-guess baseline at all tasks. Our findings indicate that there is substantial room for improvement in the audio FM's ability to understand and predict turn-taking events.

## 6 DISCUSSIONS AND CONCLUSIONS

Through an extensive survey of prior work, this work identifies the core turn management capabilities required to build an interactive voice assistant. To quantify the performance of existing audio FMs in these conversational aspects, we propose automated metrics corresponding to each capability, representing the *first effort* to assess the quality of turn-taking decisions made by audio FMs. Our findings offer valuable insights into the strengths and limitations of existing spoken dialogue systems. We plan to make our evaluation platform public, hoping that these proposed metrics will be adopted by future research to develop audio FMs that can engage in more natural and interactive conversations with users.

**Limitations**: ① *Scalability and Applicability*: Our evaluation protocol needs a supervised dataset to train the judge model, limiting its scalability and applicability. This model can however be trained on any spoken conversation dataset containing speaker turns, transcripts, and timestamps, which are often available even for non-English languages as discussed in Sec. A.7. ② *Modelling BackChannels*: Our current approach for identifying backchannels relies on heuristics (Sec. A.4) and may miss certain backchannels which is limitation of our model. ③ *Performance of Turn Taking Model*: The agreement between the judge labels and human judgments is below 80% for many metrics (Fig. 3). The primary contribution of this work lies in our novel and adaptable evaluation protocol, capable of integrating any turn-taking prediction model. Future efforts to improve model accuracy would further improve the protocol's reliability. ④ *Limitations of User Study*: This study currently tests only a few audio FMs (Sec. A.9), and we plan to expand our benchmarking as more audio FMs with turn-taking capabilities emerge. There is the potential for bias in our user study, as participants were primarily the authors and their research colleagues (see Sec. A.2).

## 7 REPRODUCIBILITY STATEMENT

To ensure reproducibility of our results and provide researchers with the ability to evaluate their own pre-trained audio FMs using our evaluation platform, we will publicly release our full codebase as part of the ESPnet (Watanabe et al., 2018) toolkit. The release can be followed here: `https://github.com/espnet/espnet/pull/5948`. Specifically, we provide: ① The code to train our judge turn-taking model (Sec. 4.1). ② The inference code for our judge model to generate turn-taking pseudo-labels $l_i$ for each chunk $i$. ③ The evaluation platform to compute our proposed metrics (Sec. 4.3) using the pseudo-labels from our judge model. Researchers can then use our evaluation platform to benchmark their own audio FMs by collecting human-AI conversation data and comparing performance against other models using our evaluation metrics.

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

# A  APPENDIX

## A.1  PRIOR WORKS ON PREDICTING TURN-TAKING EVENTS

There have been a lot of prior efforts in predicting turn change, backchannel, and interruptions from spoken conversations. Prior works (Gravano & Hirschberg, 2012; 2011) have shown that spoken conversations consist of *Transition-relevant place* (TRP), where a turn-shift usually occurs. This suggests that the listener is likely to speak up at particular places in conversation, i.e., during or after the speech, with certain acoustic and prosodic properties. Gravano & Hirschberg further, introduces a detailed annotation scheme to label various turn-taking events naturally occurring in spoken conversation and releases a Columbia Games corpus, which contains approximately 10 hours of annotated speech using this scheme.

**Predicting turn change**: Prior works (Masumura et al., 2018; Ekstedt & Skantze, 2020; da Silva Morais et al., 2023) have focused on predicting end-of-turn (EOT) events to help voice assistants determine when to speak. These studies aim to identify, based on the conversation history, whether the current speaker's utterance has concluded. Consequently, their primary objective is to distinguish between gaps and pauses, and most of these models are not designed to handle overlapping speech. Another recent work (Li et al., 2022) focuses on predicting the appropriate timing for when the assistant should speak up using acoustic and prosodic features.

Another related line of research involves detecting speaker changes in spoken conversations (Zhao et al., 2023b;a), typically assuming minimal or no overlaps. In this approach, a special speaker change token is inserted between different speakers' transcripts to generate training targets. Recent works (Fujita et al., 2019; Cornell et al., 2023) have extended this formulation by predicting the exact timestamps when a given speaker is active, a task referred to as speaker diarization. Some studies (Cornell et al., 2023) have also attempted to model speech recognition and speaker diarization jointly. However, unlike these approaches, a turn-taking prediction model must operate causally i.e. predicting whether the speaker should speak up *now* based solely on the conversation history.

**Predicting backchannels and interruptions**: Most early works (Fujie et al., 2005) aim to predict backchannel only from acoustic information. They show that acoustic features and prosody are more beneficial for knowing when to generate backchannel than the actual content of speech. Building on these findings, there has similarly been interest in predicting backchannel from multimodal information (Morency et al., 2010; Morency, 2010; Scherer et al., 2012). Similarly, efforts have been made to predict interruptions and their types (competitive/cooperative) by manually annotating the IEMOCAP dataset (Lee & Narayanan, 2010; Lee et al., 2008).

**Multi-task prediction of Turn-Taking Events**: Recent works often adopt a multi-task learning approach to jointly predict various turn-taking events. For example, one study (Hara et al., 2018) uses multitask learning to predict turn changes along with backchannels and fillers. Other studies (Ishii et al., 2021; 2020) explore joint modeling of turn changes, turn management willingness, and backchanneling. Turn management willingness is related to the mental states of the speaker and listener in spontaneous conversations and can be categorized as follows: (1) turn-holding — the speaker's willingness to continue speaking, (2) turn-yielding — the speaker's willingness to listen, (3) turn-grabbing — the listener's willingness to start speaking, and (4) listening — the listener's willingness to continue listening.

While these works required supervised labeled data, recent works like the Voice Activity Projection (VAP) models (Ekstedt & Skantze, 2022a; Inoue et al., 2024) are trained in an unsupervised manner on spoken dialogue data to predict upcoming "speech" activity for each speaker in a future time window using pre-train speech representations and voice activity detection (VAD) features. The latest version of VAP (Inoue et al., 2024) no longer requires VAD features and is multilingual. The outputs from the two channels are fed into a cross-attention Transformer, similar to the Dialog GSLM model (Nguyen et al., 2023), and the system performs multitask learning on both VAP and VAD. They then successfully trained turn-taking models on Chinese (Mandarin) and Japanese using publicly available datasets (Liu et al., 2006; Inaba et al., 2022). Interestingly, multilingual models trained on English, Chinese, and Japanese perform comparably to monolingual models despite the diverse turn-taking behaviors of these languages. Ekstedt & Skantze investigate the role of prosody in turn-taking using the VAP model

**Multimodal FM:** Recently multimodal FMs (Défossez et al., 2024; Xie & Wu, 2024) have been proposed that claim to be able to engage in real-time conversation with a user. Moshi (Défossez et al., 2024) is a full duplex E2E spoken dialogue framework based on a text language model backbone and models separately its own speech and that of the user into parallel streams. Further, Moshi does not explicitly model speaker turns, i.e., it is listening and generating audio tokens at all time, it simply initiates speaking when it predicts the special token **EPAD** and it stays silent when it predicts the special token **PAD**.

### A.1.1    PRIOR TURN TAKING EVALUATION METRICS

Most prior works (Masumura et al., 2018; Wang et al., 2024) evaluate turn-taking models based on their ability to *predict whether a turn-taking event, such as a turn change or backchannel, will occur in the near future*. This involves assessing how well turn-taking model forecasts when specific turn-taking events will happen in *human-human conversations* based on contextual cues. VAP (Ekstedt & Skantze, 2022a) similarly evaluates turn-taking models using four key metrics: (a) SHIFT vs. HOLD (S/H), assessing how well a model predicts whether the current speaker will hold the turn or whether the turn will shift to the other speaker during mutual silence; (b) SHIFT prediction, evaluating the ability to predict an upcoming speaker change, while a speaker is still active; (c) Backchannel prediction, measuring how well upcoming backchannels are predicted; and (d) SHORT vs. LONG (S/L), evaluating whether a speaker change is part of a backchannel or a proper turn change. Additionally, Ekstedt et al. propose an automatic evaluation method based on VAP to measure turn management behaviors (e.g., hold/yield) for conversational speech synthesis.

Our work distinguishes itself from these prior evaluation metrics as the *first effort to evaluate AI dialogue systems' ability to perform turn-taking events* in spontaneous, interactive conversations with human users. This involves assessing how well the AI system actively decides to take the conversation floor, yield it's turn, backchannel or interrupt the user during live human-AI interactions, reflecting its capability to engage in natural dialogue with the end user. Due to this fundamental difference in application focus, several challenges arise when attempting to apply prior turn-taking evaluation metrics (Ekstedt & Skantze, 2022a) to our setting:

① **Absence of Ground Truth**: In prior turn-taking evaluation methods, human-human turn-taking decisions serve as ground truth to assess how accurately a model predicts upcoming turn-taking events. However, in human-AI conversations, there is no inherent ground truth to evaluate the quality of AI's turn-taking decisions. Ground truth must be generated through human relevance judgments, requiring annotators to listen to entire human-AI conversations and determine whether the AI's turn-taking decisions were appropriate. For example, Ekstedt & Skantze 's SHIFT vs. HOLD (S/H) metric evaluates whether a turn-taking model can predict whether the current speaker will hold or yield its turn. We adapted this to assess whether an audio FM can *correctly* decide when to speak or when to allow the user to continue when the user pauses in interactive human-AI conversation. Since no predefined ground truth exists to judge these decisions, we introduced a protocol to generate pseudo ground-truth labels using a judge turn-taking model trained on human-human conversations. The agreement between the AI's decisions and these pseudo labels serves as a measure of the quality of the AI system's turn-taking behavior.

② **Dual Metrics for Human and AI Turns**: Unlike prior works, our evaluation protocol requires separate modeling of turn-taking decisions for when the AI is the listener (human's turn) versus when the AI is the speaker (AI's turn). For example, in addition to evaluating the AI's behavior as a listener, we also assess it based on the turn taking events made by the user when it pauses during its turn. Specifically, we examine whether the AI provides clear cues to the end user to convey its willingness to yield the floor or retain it. This distinction is crucial for understanding the AI's ability to appropriately formulate it's output to manage turn-taking effectively.

③ **Distinct Metrics for Human and AI Turns**: It is important to recognize that the same set of metrics cannot be applied uniformly to both the AI and human turns. For instance, during the human's turn, we evaluate the AI's decision to interrupt—determining whether its interruptions are relevant and timely or overly aggressive. Conversely, during the AI's turn, it is not meaningful to assess the appropriateness of user interruptions. Instead, we focus on how the AI responds to user interruptions. Specifically, we evaluate whether the AI ignores interruptions entirely or, conversely, always becomes silent, even when the user is merely providing supportive feedback rather than attempting to take over the conversation. To address these nuances of what metrics are appropriate,

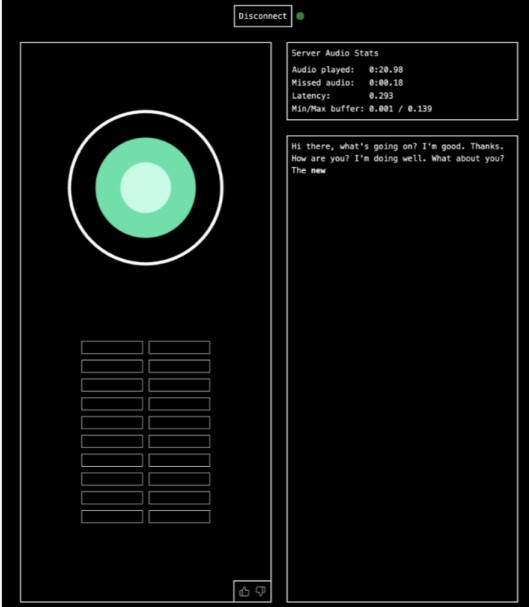

Figure 5: Screenshot of Moshi demo as shown to the participants during user study

we conducted a thorough survey of prior work (see Sec. 4.2) to identify the key turn-taking abilities required for a conversational agent to engage effectively with end users. Based on these findings, we designed appropriate metrics for each ability, adapting existing metrics when feasible and creating new ones where necessary.

④ **Introducing New Metrics**: Due to the difference in application focus, existing metrics could not be fully adapted to capture the complete range of conversational capabilities. For example, prior metrics did not address the handling of interruptions. In this work, we designed a labeling sequence for our judge model to explicitly differentiate between floor-taking (successful interruption) and butting-in (unsuccessful interruption) interruptions. This enhancement ensures that our evaluation protocol comprehensively assesses all key aspects of human-AI turn-taking behavior.

In summary, our protocol builds on prior work by identifying core turn-taking abilities for human-AI interaction, adapting existing metrics, and introducing new ones for comprehensive evaluation across all turn-taking abilities.

## A.2 USER STUDY

As discussed in Sec. 3, we run a user study where we hire participants to have a conversation with spoken dialogue systems. For Moshi, we use their publicly available demo webpage `https://moshi.chat/` for participants to converse with the dialogue systems and collect their conversation recordings. The screenshot of Moshi's demo is shown in Fig. 5. For the cascaded dialogue system, since the publicly available implementation `https://github.com/huggingface/speech-to-speech` could only be run on the command line terminal, we built a gradio demo to facilitate a more user-friendly interface for participants. A screenshot of the demo has been shown in Fig. 6, and the demo will be made publicly available upon acceptance. We collect exactly 4 hours 5 minutes and 3 hours 35 minutes of spoken conversation data across 11 different speakers with Moshi and cascaded dialogue system, respectively. The participants primarily included the authors and their research colleagues. Similar *lab experiments* are common in prior work (Deriu et al., 2021), enabling focused testing of specific functionalities (eg. prompting participants to interrupt the AI system) and facilitating deeper insights through discussions about their subjective experiences. However, we acknowledge that this controlled setup may introduce bias. We hope our evaluation protocol inspires future studies to conduct larger-scale evaluations with more diverse participants. The instructions provided to the participants are shown below:

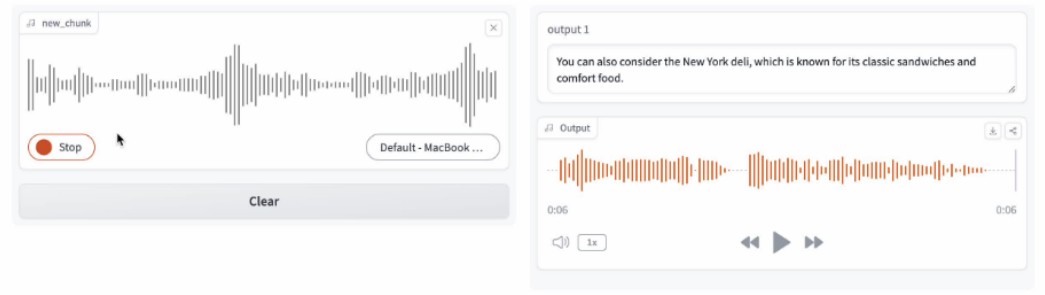

Figure 6: Screenshot of Cascaded demo as shown to the participants during user study

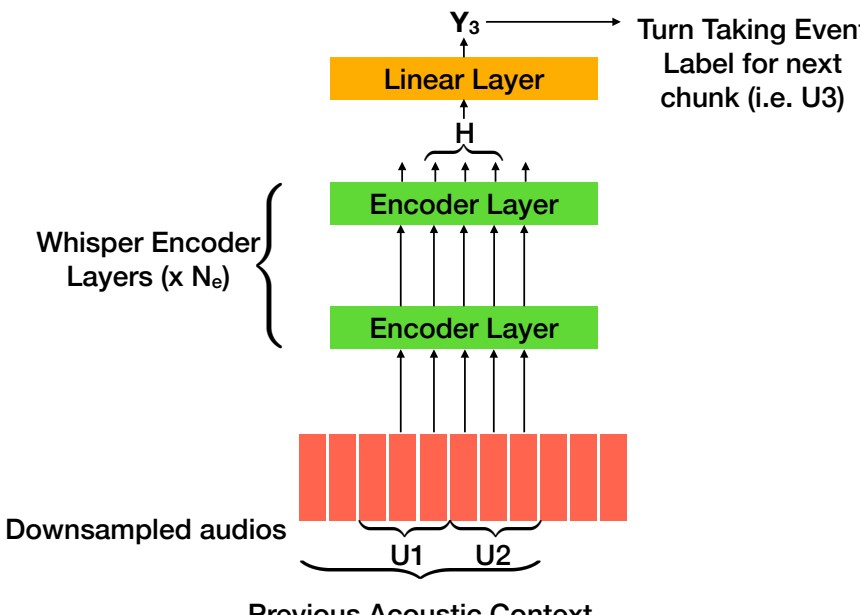

Figure 7: Architecture of our causal turn-taking model predicting at every 40ms chunk

**Instruction for User Study:** We are interested in studying and evaluating real-time dialogue interactions during human-AI (audio FMs) conversations. To this end, we recruit participants to converse with the dialogue system and will analyze your conversation recordings to gain insights on the dialogue interactions. If you are interested, please follow these guidelines to have a short conversation session ( 5 minutes) on one of these topics:
(1) Feedback on your pronunciation.
(2) Good places to eat in Manhattan.
(3) Sightseeing in New York.
(4) Ask Moshi for help in learning python.
(5) Ask Moshi for help in preparing to go for a hike.
(6) Any topic of your choice.
Please label the audio with the topic you choose. Please do not discuss anything you are not comfortable to share.

| Turn-Taking Label | Switchboard ROC-AUC | F1 |
|---|---|---|
| **C** (Continuation) | 93.1 | 84.1 |
| **BC** (Backchannel) | 89.1 | 68.6 |
| **T** (Turn change) | 90.3 | 54.8 |
| **I** (Interruption) | 91.0 | 70.2 |
| **NA** | 95.2 | 86.6 |
| Overall | 91.7 | 72.9 |

Table 5: Validation performance of supervised turn-taking prediction model (Sec. 4.1).

### A.3 ARCHITECTURE OF JUDGE TURN-TAKING MODEL

The input to the model is single-channel mixed speech $X$ which can be represented as a $T$-length sequence of features $X = \{\mathbf{x}_t | t = 1, \ldots, T\}$. Through the maximum a posteriori (MAP) decision theory, our turn-taking model estimates the label sequence $L$ by maximizing posterior probability $P(L|X)$. By product rule, we have $P(L|X) = \prod_{i=1}^{N} P(l_i|l_{1:i-1}, X)$. For simplicity, we divide the entire conversation audio into a sequence of N non-overlapping chunks $U = \{U_i \mid i = 1, \ldots, N\}$ such that $U_i = \mathbf{x}_{B_{i-1}+1:B_i}$ where $B_i = i * N_{block}$ and $N_{block}$ is the size of the chunk. Our model is *causal* such that it is trained to make future predictions, i.e., what turn-taking event will happen in the *next* chunk? Hence, we condition on only the first $i - 1$ chunks i.e., $U_{1:i-1}$:

$$P(L|X) = \prod_{i=1}^{N} P(l_i|l_{1:i-1}, U_{1:i-1}). \tag{6}$$

From Sec. 3.1, we have $U_{1:i-1} = x_{1:B_{i-1}}$. By C.I. assumption of $l_i \perp\!\!\!\perp l_{1:i-1} \mid U_{1:i-1}$, we get $P(L|X) = \prod_{i=1}^{N} P(l_i|x_{1:B_{i-1}})$. We can further simplify by conditioning only on a prior context window $\hat{U}_i$ of size $W$:

$$\hat{U}_i = x_{(B_{i-1}-W):B_{i-1}}. \tag{7}$$

Using $\hat{U}_i$ and CI assumption, we simplify Eq. 6 to:

$$P(L|X) = \prod_{i=1}^{N} P(l_i|\hat{U}_i). \tag{8}$$

Fig. 7 shows the architecture of our causal turn-taking model. The input speech is first passed through encoders (Encoder($\cdot$)) of the pre-trained speech foundation model, namely Whisper (Radford et al., 2022). We perform a weighted sum of the encoder's hidden states (Yang et al., 2021) and then use the encoder output of the last audio frame as acoustic representation $\mathbf{h}_i = \text{Encoder}(\hat{U}_i)$. The acoustic representations are then passed through linear layer (Out($\cdot$)) followed by softmax:

$$\mathbf{f}(\hat{U}_i, \theta) = \text{Softmax}(\text{Out}(\mathbf{h}_i)), \tag{9}$$

where $\mathbf{f}(\hat{U}_i, \theta)[l_i]$ is the likelihood that the turn-taking event $l_i$ will happen at chunk $i$ as predicted by our supervised turn-taking model. The entire model is then trained using cross entropy loss. It is important to note that while some prior approaches (Ekstedt & Skantze, 2022a) can model interruptions, they require separate channels for each speaker, which might not always be available. Our approach, however, can effectively model both speakers' turn-taking behavior using single-channel mixed speech, thereby overcoming this limitation.

### A.4 EXPERIMENT SETUP OF SUPERVISED TURN-TAKING MODEL

The size of the chunk, i.e., $N_{block}$ is 40msec. We use the Whisper medium encoder to generate acoustic representations. The context window of the supervised turn-taking model $W$ is 30 seconds. To get backchannel annotations, we follow Wang et al. and use the most common isolated one and two-word phrases as backchannels [6]. Following (Wang et al., 2024), we downsample chunks with

---

[6]`https://github.com/ErikEkstedt/VoiceActivityProjection/blob/main/dataset_swb/backchannels.csv`

| | Switchboard | |
| Turn-Taking Metric | Agreement | Margin of Error |
| --- | --- | --- |
| When to speak up? | | |
|     When listener decides to speak up | 81.2 | ±6.43 |
|     When listener lets speaker continue | 75.5 | ±7.46 |
| When to backchannel? | | |
|     When listener backchannels | 71.6 | ±5.75 |
|     When listener does not backchannels | 74.8 | ±3.49 |
| When to interrupt? | | |
|     When listener interrupt | 76.6 | ±5.13 |
|     When listener does not interrupt | 72.8 | ±1.80 |
| Handle user interruptions? | | |
|     When interrupting speaker takes the floor | 61.3 | ±5.90 |
|     When interrupting speaker does not take the floor | 57.0 | ±5.30 |

Table 6: Results presenting alignment of judge label with human decisions in in-domain validation set. We also show the Margin of Error (ME) (Tanur, 2011) for 95% confidence intervals to quantify uncertainty in agreement between judge labels and human relevance judgments due to sensitivity to threshold values as discussed in Sec. A.8.

| Turn-Taking Metric | Moshi | Cascaded |
| --- | --- | --- |
| (a) When to speak up? | | |
|     When listener decides to speak up | [88.7, 94.2] | [84.2, 88.9] |
|     When listener lets speaker continue | [48.9, 56.5] | [38.4, 48.3] |
| (b) When to backchannel? | | |
|     When listener backchannels | [22.9, 63.4] | [ 0.0, 59.5] |
|     When listener does not backchannels | [85.7, 87.8] | [90.7, 92.5] |
| (c) When to interrupt? | | |
|     When listener interrupt | [31.1, 42.0] | [11.8, 32.6] |
|     When listener does not interrupt | [77.3, 80.6] | [83.9, 85.5] |
| (d) Convey user to speak up? | | |
|     When listener decides to speak up | [79.0, 84.9] | [77.3, 86.2] |
|     When listener lets speaker continue | [24.9, 36.2] | [36.9, 44.1] |
| (e) Handle user interruptions? | | |
|     When interrupting speaker takes the floor | [26.2, 84.0] | [ 0.0, 35.3] |
|     When interrupting speaker does not take the floor | [37.2, 65.2] | [89.2, 100.0] |

Table 7: Confidence intervals for the consistency of the AI dialogue system's turn-taking decisions with judge labels across our proposed metrics.

| Dialogue System | Speaking Rate | Backchannel rate |
|---|---|---|
| Switchboard | 204.5 | 4.00 |
| Moshi | 223.7 | 0.25 |
| Cascaded | 206.5 | 0.13 |

Table 8: Performance of audio foundation models on engaging in conversation with humans based on Natural Dialogue Event Statistics i.e. Speaking rate and Backchannel rate.

label class **C**, **NA**, **BC** and **I** from training set such that there are roughly similar numbers of samples for each label class. The validation performance is shown in Table 5.

We evaluated our model's OOD generalization on two out-of-domain (OOD) datasets: (1) the Columbia Games Corpus (Gravano & Hirschberg, 2011), a 10-hour task-oriented spoken dialog corpus, and (2) the Fisher Corpus (Cieri et al., 2004), a non-task-oriented spoken dialog corpus. The Fisher dataset's transcriptions were created using the Quick Transcription specification, which introduced inaccuracies and left significant portions untranscribed. To address this, we developed heuristics to identify and exclude audio segments with large untranscribed content from the test set. We created a random test split of 138 conversations, comprising 23 hours of audio, which we will make publicly available. Further manual analysis revealed errors in the ground truth timestamps. We corrected these timestamps using speaker diarization outputs from Pyannote. All data preparation code and our turn-taking model will also be made publicly available.

For our proposed metrics (Sec. 4.4-4.8), we get the following values for threshold i.e. $\text{threshold}_1 = 0$, $\text{threshold}_2 = 0.1$, $\text{threshold}_3 = -0.45$, $\text{threshold}_4 = -0.1$. The agreement between the judge labels and human judgment on the in-domain validation set is shown in Table 6. We discuss the impact of threshold tuning on our judge model's agreement with human judgment in Sec. A.8.

For single-label evaluations (Sec. 4.9), operating points or thresholds for the predicted likelihood of label **C** = 0.2, **NA** = 0.45, **I** = 0.4, **BC** = 0.4, **T** = 0.4. The validation F1 using these thresholds is shown in Tab. 5.

### A.5 DATASET AND AUDIO FM DETAILS FOR ADDITIONAL EVALUATION

**Understand Turn-Taking Events**: To evaluate this, we create a test bed where positive samples are those where $\exists i : l_i = \mathbf{l}$ and negative samples are those where $\forall i : l_i \neq \mathbf{l}$ where $\mathbf{l}$ is either **T**, **BC** or **I**. The audio FM is then prompted to answer a simple Yes/No question. We create a test benchmark from 1500 samples from Switchboard's test set, such that there are 750 positive and 750 negative samples. Each audio sample has a maximum length of 10 seconds. We evaluate both open-source Audio FMs, such as Qwen-Audio-Chat (Chu et al., 2023) (7B), Qwen2-Audio-Instruct (Chu et al., 2024) (7B), and SALMONN (Tang et al., 2024) (13B), as well as a cascade system combining Whisper and GPT-4o using the OpenAI API. We manually construct five prompts to query these FMs and present the results using the most effective prompt. The best prompt for each audio FM is detailed in Tab. 12.

**Predict Turn-Taking Events**: The samples in this evaluation benchmark for predicting turn change, backchannel, interruption, and floor-taking interruption are obtained from instances used to compute human relevance judgments for Metrics **(1)**, **(2)**, **(3)**, and **(5)**, respectively. Similar to Setting 1, the audio FM is again prompted to answer a simple Yes/No question. Similar to before, each evaluation benchmark consist of 1500 samples[7] with equal positive and negative instances. Each audio sample has a maximum length of $W = 30$ seconds, matching the context window of the supervised topline model. We evaluate the same audio FMs as before. We again manually construct five prompts to query these FMs and the best prompt for each audio FM is detailed in Tab. 13 and 14.

### A.6 NATURAL DIALOGUE EVENT STATISTICS

The input speech for each utterance is passed through an ASR model (Radford et al., 2022) to get the corresponding ASR transcript. We use the generated transcript to get the speaking rate (i.e.

---

[7]Except benchmark of floor-taking interruptions, which only has 250 samples since interruptions are rare.

number of words/minute). We compute backchannel sequence $BC^{AI}$ and $BC^{Human}$ (See Sec. 4.3) and report the backchannel word rate (i.e. number of backchannel words/minute) Tab. 8 shows that ① both dialogue systems speak faster than the average human, with Moshi, in particular, speaking at a significantly higher rate, and ② both dialogue systems do not produce a similar number of backchannels as humans.

### A.7 Scalability and Applicability to non English and low reseource Languages

As noted in Appendix A.1, prior work (Inoue et al., 2024) successfully trained turn-taking models on Chinese (Mandarin) and Japanese using publicly available datasets. Interestingly, multilingual models trained on English, Chinese, and Japanese perform comparably to monolingual models despite the diverse turn-taking behaviors of these languages.

To address scenarios without supervised datasets, we propose a low-cost solution: collecting a small spoken dataset for the target language and generating annotations through human efforts or using tools like PyAnnote (speaker diarization) and Whisper (ASR). We can then train multilingual turn-taking models that leverage high-resource language data to improve performance on low-resource languages. This approach is still far more cost-effective than collecting human relevance judgments for every turn-taking event.

### A.8 Impact of tuning thresholds for judge labels

The number of turn taking events are extremely unbalanced with continuation (C) and silence (NA) making up the label set for more than 95% of instances. As a result, it is a common practice in prior literature (Ekstedt & Skantze, 2022a) to tune the threshold for each label on the validation set. We further surveyed prior literature to better quantify the impact of threshold tuning on our judge model's performance and identified two main approaches:

① Sensitivity Analysis (Novello et al., 2022; Razavi et al., 2021; Sadeghi & Matwin, 2024): Prior works have experimented with varying key hyperparameters systematically over a range and analyzed validation performance as a function of hyperparameters. Inspired by this, we vary thresholds from -0.5 to 0.5 for metrics where judge labels are computed using the difference of 2 likelihoods (all metrics (Sec. 4.4- 4.8) except metric (B) (see Eq. 3)) and from 0 to 1 for others (i.e., for metric B in Sec. 4.5). The thresholds are incremented in steps of 0.01. We then calculated the Margin of Error (ME) (Tanur, 2011) for 95% Confidence Intervals to quantify uncertainty in agreement between judge labels and human relevance judgments due to sensitivity to threshold values. Margin of error (ME) is calculated as:

$$\text{ME} = z \cdot \frac{\sigma}{\sqrt{n}} \tag{10}$$

Where (z = 1.96): Z-score for a 95% confidence level, ($\sigma$): Standard deviation and (n): Sample size (i.e. Size of threshold range over which performance is computed =100 for our experiments)
The observed margin of errors, along with the agreement of the judge label with human decisions on the in-domain validation set, are shown in Tab. 6. Generally, a margin of error of below 5% is considered excellent for high-accuracy needs, and less than 10% is acceptable for most studies. Our analysis shows that the agreement with human judgements does not undergo huge fluctuations with changes in threshold (i.e., margin of error is always less than 10%), and hence, our approach does not require extensive hyperparameter tuning.

② Validation on Multiple Datasets (van Rijn & Hutter, 2018): Prior works have argued the reliability of their model by showing consistent performance across datasets, demonstrating that the model generalizes well without dataset-specific tuning. In Figure 3, our judge labels achieve good agreement with human judgment even on the OOD spoken dialog dataset, i.e., Columbia Games Corpus, without a threshold being specifically tuned for this dataset. This result shows that our model achieves consistent performance without dataset-specific tuning.

### A.9 Results with GPT-4o

When GPT-4o's Advanced Voice Mode became publicly available, we initiated data collection of human-GPT-4o conversations, accumulating 2 hours and 36 minutes of audio across six speakers. Our experiments have revealed many interesting insights.

| Turn taking event | Number of events/minute | % Cumulated Duration |
|---|---|---|
| IPU | 13.9 | 71.9 |
| Pause | 11.4 | 13.2 |
| Gap | 2.3 | 15.2 |
| Overlap | 0.3 | 0.2 |

Table 9: Results of GPT-4o on engaging in conversation with humans based on corpus-level statistics proposed in prior works (Nguyen et al., 2023).

| AI is Listener | | | AI is Speaker | |
|---|---|---|---|---|
| (a) % Turn Change | (b) % Backchannel | (c) % Interruption | (d) % Turn Change | % Interruption ((e) % Floor-taking) |
| 37.2 | 0.003 | 0.06 | 13.6 | 0.04 (67.3) |

Table 10: Percentage of instances for each proposed metric where GPT-4o makes a specific turn-taking decision. Brackets in the last column show the percentage of interruptions that result in floor-taking.

**Corpus-level Statistics**: The corpus level statistics for human GPT-4o conversation are shown in Tab. 9. ① Turn-Taking Latency: GPT-4o exhibits a moderate gap between speaker turns (13.2% of cumulative duration in Tab. 9), smaller than the Cascaded system (32.5% in Fig. 2b) but larger than Moshi (11.8% in Fig. 2b), indicating intermediate latency. ② Overlap: Similar to a cascaded system (Fig. 2b), GPT-4o has minimal overlapping speech (0.2% of cumulative duration in Tab. 9), resulting in less interactive conversations. ③ Turn-Yielding Behavior: GPT-4o has a high number of pause events (11.4 per minute) and fewer gap events (2.3 per minute) compared to the other dialogue systems (Fig. 2a), indicating that it sometimes speaks for very long without yielding it's turn, which makes the conversation bland and less engaging for end users.

**Our Proposed Metrics**: ① Tab. 10 show that instances of GPT-4o backchanneling and GPT-4o and user interrupting each other are too few. This leads to large confidence intervals for Metrics B, C and E, making it challenging to derive conclusive insights. ② Metric A: GPT-4o demonstrates strong performance in determining when to allow the user to continue speaking, achieving 71.6% agreement with human judge labels (Tab. 11). This is significantly higher than Moshi (52.8%) and the Cascaded system (44.7%) (Fig. 3a). ③ Metric D: GPT-4o is similarly significantly better at conveying users when it wants to keep the conversation floor, having 70.5% agreement with judge labels (Tab. 11) which is much higher than Moshi (32.7% agreement in Fig.3d) and Cascaded system (40.8% agreement in Fig.3d).

While our analysis is limited to a few audio FMs, this is a developing field of research, and we plan to expand our benchmarking as more audio FMs with turn-taking capabilities emerge.

| Turn-Taking Metric | % Consistency with Judge | Confidence Interval |
|---|---|---|
| (a) When to speak up? | | |
|     When listener decides to speak up | 84.8 | [ 70.8, 89.0] |
|     When listener lets speaker continue | 71.6 | [ 62.9, 81.4] |
| (b) When to backchannel? | | |
|     When listener backchannels | 0.0 | [ 0.0, 0.0] |
|     When listener does not backchannels | 89.3 | [ 86.4, 90.7] |
| (c) When to interrupt? | | |
|     When listener interrupt | 59.3 | [-12.7, 58.3] |
|     When listener does not interrupt | 88.2 | [ 85.9, 90.6] |
| (d) Convey user to speak up? | | |
|     When listener decides to speak up | 86.9 | [ 80.6, 91.4] |
|     When listener lets speaker continue | 70.5 | [ 60.8, 75.2] |
| (e) Handle user interruptions? | | |
|     When interrupting speaker takes the floor | 51.4 | [ 26.2, 83.8] |
|     When interrupting speaker does not take the floor | 70.6 | [ 14.1, 93.5] |

Table 11: Consistency of the GPT-4o's turn-taking decisions with judge labels across our proposed metrics along with 95% confidence intervals

| Turn-taking event | Model | Prompts | Example Output |
|---|---|---|---|
| Turn Change | SALMONN | | No, the given audio does not have a turn change. |
| | Qwen2-Audio-Instruct | Does the given audio has a turn change anywhere? Briefly explain your answer. You must reply with (a) Yes or (b) No. | No, there is no turn change in the speech as the pitch and tone remain consistent throughout the recording. |
| | Qwen-Audio-Chat | | Yes, there is a turn change in the given audio. The male speaker starts speaking and then the female speaker starts speaking. |
| | Whisper+GPT-4o | Does the given audio has a turn change anywhere? Briefly explain your answer. You must reply with (a) Yes or (b) No. | (b) No |
| Backchannel | SALMONN | Backchannels are brief words or sounds (like 'um,' 'right,' or 'okay') that a listener makes to show they're paying attention. Do the 2 speakers backchannel anywhere in the provided audio? Briefly explain your answer. You must reply with (a) Yes or (b) No. | No, the speakers do not backchannel in the provided audio. |
| | Qwen2-Audio-Instruct | | No, there are no backchannels in this audio. |
| | Qwen-Audio-Chat | | Yes, the two speakers backchannel briefly in the provided audio. |
| | Whisper+GPT-4o | Do the 2 speakers backchannel anywhere in the provided audio? Briefly explain your answer. You must reply with (a) Yes or (b) No. | (b) No |
| Interruption | SALMONN | Do the 2 speakers interrupt each other anywhere in the provided audio? Briefly explain your answer. You must reply with (a) Yes or (b) No. | No, the 2 speakers do not interrupt each other in the provided audio. |
| | Qwen2-Audio-Instruct | | No, the speakers do not interrupt each other. They speak at different times and their overlapping is minimal. |
| | Qwen-Audio-Chat | | Yes, the two speakers interrupt each other. |
| | Whisper+GPT-4o | Do the 2 speakers interrupt each other anywhere in the provided audio? Briefly explain your answer. You must reply with (a) Yes or (b) No. | (b) No |

Table 12: Prompts for audio foundation models on test benchmarks evaluating their ability to *understand* turn-taking events. All open source FMs have same best prompt.

| Turn-taking event | Model | Prompts | Example Output |
|---|---|---|---|
| Turn Change | SALMONN Qwen2-Audio-Instruct Qwen-Audio-Chat | You are provided the audio of a 2 speaker conversation. The current speaker takes pause at the end of current audio. Given the provided audio, do you think that after the pause, the current speaker will (A) continue talking or (B) the second speaker will speak up. Output only A or B, nothing else. Among A or B, the answer is | (B) The second speaker will speak up. The current speaker will continue talking.

B |
| | Whisper+GPT-4o | You are provided the audio of a 2 speaker conversation. The current speaker takes pause at the end of current audio. Given the provided audio, do you think that after the pause, the current speaker will (a) continue talking or (b) the second speaker will speak up. Briefly explain your answer. You must reply with one of 2 options. | (b) the second speaker will speak up. The current speaker has asked a direct question, which typically prompts a response from the second speaker. |
| Backchannel | SALMONN

Qwen2-Audio-Instruct Qwen-Audio-Chat | You are given the audio of a two-speaker conversation. Based on the provided audio, do you predict that either speaker will produce a backchannel (e.g., a short acknowledgment or response) immediately after the end of the audio? Please explain your reasoning briefly. You must answer (a) Yes or (b) No. | No, I do not predict that either speaker will produce a backchannel immediately after the end of the audio.
Yes

Based on the provided audio, it is likely that either speaker will produce a backchannel response immediately after the end of the audio. This is because the conversation is between two people who are discussing a topic and exchanging ideas, and it is common for people to respond or acknowledge each other after a pause or when the conversation is wrapping up. |
| | Whisper+GPT-4o | You are provided the audio of a 2 speaker conversation. Given the provided audio, do you think any of the 2 speaker will backchannel exactly after the end of the audio. Answer as (a) yes or (b) no. Briefly explain your answer. | (b) no. Explanation: The provided text does not indicate any clear cues or signals that would prompt an immediate backchannel response from the other speaker. The speaker is expressing personal thoughts and opinions, and there is no evident pause or question that would naturally lead to a backchannel response right at the end of the audio. |
| Interruption | SALMONN

Qwen2-Audio-Instruct

Qwen-Audio-Chat | During spontaneous human conversation, interruptions occur frequently and often correspond to breaks in the information flow between conversation partners. You are analyzing a brief audio clip from a two-person conversation. The clip ends abruptly while the current speaker is still in the middle of a sentence. Based ONLY on the provided audio, is it more likely that: (a) The other speaker will interrupt BEFORE the current speaker finishes their sentence? (b) The other speaker will let the current speaker to continue speaker? Justify your choice, considering the factors that might lead to an interruption or encourage someone to wait their turn. | (a) The other speaker will interrupt BEFORE the current speaker finishes their sentence. It is not possible to determine with certainty whether the other speaker will interrupt before the current speaker finishes their sentence based solely on the provided audio. (a) The other speaker will interrupt BEFORE the current speaker finishes their sentence. The conversation ends abruptly, and the other speaker is still speaking, suggesting that they were interrupted. Additionally, the other speaker mentions "that wouldn't be fair" and "what they would have to do," indicating that they may be about to make an argument or disagreement, which could lead to an interruption. |
| | Whisper+GPT-4o | During spontaneous human conversation, interruptions occur frequently and often correspond to breaks in the information flow between conversation partners. Interruptions can happen due to a variety of reasons like urgency, disagreement, etc. Interruptions can be broadly classified into two categories: competitive and cooperative interruptions. Occurrences of competitive interruptions are usually disrupting to the flow of conversation between speakers while cooperative interruptions are more supportive to the flow. You are analyzing a brief audio clip from a two-person conversation. The clip ends abruptly while the current speaker is still in the middle of a sentence. It is unclear from the provided audio whether the speaker is about to pause, is struggling to find the right word, or will continue speaking uninterrupted. Based ONLY on the provided audio, is it more likely that: (a) The other speaker will interrupt BEFORE the current speaker finishes their sentence? (b) The other speaker will let the current speaker to continue speaker? Justify your choice, considering the factors that might lead to an interruption or encourage someone to wait their turn. | Based on the provided audio, it is more likely that: (b) The other speaker will let the current speaker continue speaking. Justification: 1. **Flow of Conversation**: The current speaker is narrating a story with a clear sequence of events. The narrative seems to be reaching a conclusion, as indicated by the speaker's expression of satisfaction ("I just had the biggest grin on my face"). 2. **Listener's Engagement**: The other speaker has already shown engagement and support by interjecting with "I'll bet you did," which is a cooperative interruption. This indicates that the listener is following the story and is likely to allow the speaker to finish their thought. 3. **Lack of Urgency or Disagreement**: There are no signs of urgency or disagreement in the provided audio. The tone of the conversation is casual and friendly, which reduces the likelihood of a competitive interruption. 4. **Narrative Completion**: The current speaker is in the middle of a sentence that seems to be wrapping up the story. Interrupting at this point would disrupt the natural conclusion of the narrative. Given these factors, it is more reasonable to expect that the other speaker will wait for the current speaker to finish their sentence before responding. |

Table 13: Prompts for audio foundation models on test benchmarks evaluating their ability to *predict* turn-taking events. All open source FMs have same best prompt.

| Turn-taking event | Model | Prompts | Example Output |
|---|---|---|---|
| Successful | SALMONN | You are examining a short audio clip from a conversation between two people. The audio ends suddenly RIGHT as the second speaker begins to talk, overlapping with the first speaker. It's impossible to tell from this brief excerpt if the first speaker was about to pause, was having trouble finding a word, or would have continued speaking. Based ONLY on the provided audio, is it more likely that: (A) The second speaker's overlap is a successful interruption, and they will take over the conversation? (B) The second speaker's overlap is a mistimed attempt to start their turn, and the first speaker will continue, forcing the second speaker to wait for a natural break or pause before speaking? Output only A or B, nothing else. Among A or B, the answer is | (B) The second speaker's overlap is a mistimed attempt to start their turn, and the first speaker will continue, forcing the second speaker to wait for a natural break or pause before speaking |
| | Qwen2-Audio-Instruct | | B |
| Interruption | Qwen-Audio-Chat | | Based solely on the provided audio, it is more likely that the second speaker's overlap is a successful interruption, and they will take over the conversation. This is because the audio ends abruptly as the second speaker begins to speak, suggesting that the first speaker was interrupted. Additionally, the second speaker's tone sounds confident and assertive, which is characteristic of someone who is trying to take over a conversation. On the other hand, the first speaker's response is short and does not indicate that they are about to pause or continue speaking. |
| | Whisper+GPT-4o | During spontaneous human conversation, interruptions occur frequently and often correspond to breaks in the information flow between conversation partners. Interruptions can happen due to a variety of reasons like urgency, disagreement, etc. Interruptions can be broadly classified into two categories: competitive and cooperative interruptions. Occurrences of competitive interruptions are usually disrupting to the flow of conversation between speakers while cooperative interruptions are more supportive to the flow. You are examining a short audio clip from a conversation between two people. The audio ends suddenly RIGHT as the second speaker begins to talk, overlapping with the first speaker. It's impossible to tell from this brief excerpt if the first speaker was about to pause, was having trouble finding a word, or would have continued speaking. Based ONLY on the provided audio, is it more likely that: (a) The second speaker's overlap is a successful interruption, and they will take over the conversation? (b) The second speaker's overlap is a mistimed attempt to start their turn, and the first speaker will continue, forcing the second speaker to wait for a natural break or pause before speaking? Justify your choice, considering the factors that might lead to a successful interruption or encourage someone to wait their turn, but focus on how the SPECIFIC TIMING of this overlap makes one outcome more likely than the other. | Based on the provided audio excerpt, it is more likely that: (b) The second speaker's overlap is a mistimed attempt to start their turn, and the first speaker will continue, forcing the second speaker to wait for a natural break or pause before speaking. Justification: 1. **Context and Flow**: The first speaker is in the middle of a detailed explanation about weather patterns and their impact on snowfall. They have not yet reached a clear conclusion or natural pause in their narrative. The sentence B̈ut I have playeds̈uggests that the first speaker is about to continue with a related point or anecdote. 2. **Timing of Overlap**: The overlap occurs just as the first speaker is transitioning to a new but related point (B̈ut I have played). This timing suggests that the first speaker is not finished and is likely to continue speaking. The second speaker's attempt to interject at this moment is mistimed because it interrupts the flow of the first speaker's ongoing thought. 3. **Conversational Dynamics**: In spontaneous conversation, speakers often signal the end of their turn with a clear pause or a concluding statement. The first speaker has not provided such a signal, making it more likely that they will continue speaking despite the second speaker's attempt to interject. Given these factors, the specific timing of the overlap makes it more likely that the first speaker will continue, and the second speaker will need to wait for a more appropriate moment to take their turn. |

Table 14: Prompts for audio foundation models on test benchmarks evaluating their ability to *predict* turn-taking events. All open source FMs have same best prompt.

