# OpenReview forum: "Talking Turns: Benchmarking Audio Foundation Models on Turn-Taking Dynamics"
_ICLR.cc/2025/Conference — ICLR 2025 Poster_

### Official Review · Reviewer_mo78 · 2024-11-03

**Soundness:** 3
**Presentation:** 3
**Contribution:** 3
**Rating:** 6
**Confidence:** 3

**Summary:**

This paper introduces a novel evaluation framework for assessing turn-taking capabilities in audio foundation models (FMs). The authors first propose metrics for five core conversational abilities: determining when to speak up, backchannel, interrupt, convey turn-taking cues, and handle interruptions. They develop a supervised model trained on human-human conversations to serve as a judge for evaluating these turn-taking events. Using this framework, they conducted a user study with different spoken dialogue systems (full-duplex E2E spoken dialogue system Moshi and VAD-based cascade dialogue system) and evaluated them. They evaluate several open-source and proprietary audio FMs on their ability to understand and predict turn-taking events.

**Strengths:**

1. The evaluation protocol is novel and well-motivated.
2. The experimental analysis provides valuable insights into turn-taking capabilities of audio foundation models (FMs).
3. The user study reveals noteworthy observations about current spoken dialogue systems.

**Weaknesses:**

1. Turn-taking prediction models used in evaluation protocol require training, which limits scalability and applicability.
2. The paper does not thoroughly address how its proposed evaluation protocol compares with previous turn-taking approaches, such as Ekstedt and Skantze (2022).

Reference
* Ekstedt, Erik, and Gabriel Skantze. Voice activity projection: Self-supervised learning of turn-taking events. Interspeech 2022

**Questions:**

1. What is the main difference between the proposed evaluation protocol and the previous approach by Ekstedt and Skantze (2022)? Is it impractical to apply the metrics from prior turn-taking evaluation methods to audio FMs?
2. While the turn-taking prediction model has been evaluated on an out-of-domain task-oriented spoken dialogue corpus, could you evaluate it on additional non-task-oriented spoken dialogue datasets to assess the generalizability of the model?

---

> ### Author Response · Authors · 2024-11-21
> **Response to Reviewer mo78**
>
> Thanks for the valuable comments and acknowledging that our work offers noteworthy observations about current spoken dialogue systems. We address your concerns below.
>
> ---
>
> # 1. Generalization to non-task-oriented spoken dialogue datasets
>
> We evaluate our turn-taking prediction model on Fisher. Please check the general response.
>
> ---
>
> # 2. Difference with prior turn taking evaluation metrics
> > What is the main difference between the proposed evaluation protocol and the previous approach by Ekstedt and Skantze (2022)? Is it impractical to apply the metrics from prior turn-taking evaluation methods to audio FMs?
>
> Thank you for this thoughtful question, and we apologize if this distinction was unclear in the draft. The key difference between our evaluation protocol and prior approaches [1], lies in the evaluation focus and application context.
>
> [1] focuses on assessing turn-taking model's ability to **predict turn-taking events** that will happen in the near future. This involves assessing how well turn-taking model forecasts when specific turn-taking events will happen in **human-human conversations** based on contextual cues.
>
> In contrast, our protocol evaluates AI dialogue systems' ability to **perform turn-taking events** in spontaneous, interactive conversations with human users. This involves assessing how well the AI system actively decides to take the conversation floor, yield it’s turn, backchannel or interrupt the user during live **human-AI interactions**, reflecting its capability to engage in natural dialogue with the end user. This shift required adapting existing metrics and creating new ones tailored to human-AI interaction.
>
> For example, while [1]'s SHIFT vs. HOLD (S/H) metric evaluates how well a turn-taking model predicts whether the current speaker will hold a turn or there will be a turn shift, we adapted it to evaluate an audio FM’s decision to speak or allow the user to continue during pauses. Pseudo ground-truth labels are generated using a judge turn-taking model, and the agreement between the AI's decisions and these pseudo labels is used to assess the quality of the AI system's turn-taking decisions. Additionally, we introduced novel metrics, such as evaluating how well AI systems manage user interruptions—an essential aspect of realistic human-AI interactions.
>
> In summary, our protocol represents the first effort to assess audio FMs' ability to perform turn-taking events in human-AI conversations, offering valuable insights for future research. In response to your feedback, we have added this discussion in Sec. 4.3 and A.1.1 to clarify this novel contribution.
>
> ---
>
> # 3. Scalability and Applicability of Approach
> > Turn-taking prediction models used in evaluation protocol require training, which limits scalability and applicability.
>
> Please check the general response.
>
> ---
>
> References
>
> [1] Ekstedt, Erik, and Gabriel Skantze. Voice activity projection: Self-supervised learning of turn-taking events. Interspeech 2022 (https://arxiv.org/abs/2205.09812 )

---

> ### Comment · Reviewer_mo78 · 2024-11-23
>
> Thank you for your reply and hard work. I think 1) Generalization to non-task-oriented spoken dialogue datasets is resolved with your additional experiments on Fisher and 3) Scalability and applicability is somewhat explained while it remained a limitation
> However, I still have a major concern about 2) Difference with prior turn-taking evaluation metrics: If current turn-taking metrics evaluate human-human conversation and your proposed metric evaluates human-AI interaction, isn't it possible to apply the current turn-taking metrics to human-AI interaction? In particular, could you explain what specific aspects of human-AI turn-taking cannot be captured by adapting existing metrics? Could you elaborate on this further?

---

> > ### Author Response · Authors · 2024-11-24
> > **Official Comment by Authors**
> >
> > Thank you for engaging with our response and for your thoughtful questions.
> >
> > **Recap: What is our goal?** Prior turn-taking evaluation methods only assess how well turn taking models predict future turn-taking events in human-human conversations. In contrast, our protocol evaluates the AI dialogue system's ability to actively perform turn-taking events—assessing the quality of its decisions when taking the conversation floor, backchanneling, or interrupting the user during live human-AI interactions.
> >
> > Due to this fundamental difference in application focus, several challenges arise when attempting to apply prior turn-taking metrics to our setting:
> >
> > 1. **Absence of Ground Truth**: In prior turn-taking evaluation methods, human-human turn-taking decisions serve as ground truth to assess how accurately a model predicts upcoming turn-taking events. However, in human-AI conversations, there is no inherent ground truth to evaluate the quality of AI’s turn-taking decisions. Ground truth must be generated through human relevance judgments, requiring annotators to listen to entire human-AI conversations and determine whether the AI’s turn-taking decisions were appropriate. For example, [1]'s SHIFT vs. HOLD (S/H) metric evaluates whether a turn taking model can predict whether the current speaker will hold or yield it’s turn. We adapted this to assess whether an audio FM can “correctly” decide when to speak or when to allow the user to continue when the user pauses in interactive human-AI conversation. Since no predefined ground truth exists to judge these decisions, we introduced a protocol to generate pseudo ground-truth labels using a judge turn-taking model trained on human-human conversations. The agreement between the AI’s decisions and these pseudo labels serves as a measure of the quality of the AI system's turn-taking behavior.
> > 2. **Dual Metrics for Human and AI Turns**: Unlike prior works, our evaluation protocol requires separate modeling of turn-taking decisions for when the AI is the listener (human’s turn) versus when the AI is the speaker (AI’s turn). For example, in addition to evaluating the AI's behavior as a listener, we also assess it based on the turn taking events made by the user when it pauses during its turn. Specifically, we examine whether the AI provides clear cues to the end user to convey its willingness to yield the floor or retain it. This distinction is crucial for understanding the AI’s ability to appropriately formulate it’s output to manage turn-taking effectively.
> > 3. **Distinct Metrics for Human and AI Turns**: It is important to recognize that the same set of metrics cannot be applied uniformly to both the AI and human turns. For instance, during the human’s turn, we evaluate the AI’s decision to interrupt—determining whether its interruptions are relevant and timely or overly aggressive. Conversely, during the AI’s turn, it is not meaningful to assess the appropriateness of user interruptions. Instead, we focus on how the AI responds to user interruptions. Specifically, we evaluate whether the AI ignores interruptions entirely or, conversely, always becomes silent, even when the user is merely providing supportive feedback rather than attempting to take over the conversation. To address these nuances of what metrics are appropriate, we conducted a thorough survey of prior work to identify the key turn-taking abilities required for a conversational agent to engage effectively with end users. Based on these findings, we designed appropriate metrics for each ability, adapting existing metrics when feasible and creating new ones where necessary.
> > 4. **Introducing New Metrics**: Due to the difference in application focus, existing metrics could not be fully adapted to capture the complete range of conversational capabilities. For example, prior metrics did not address the handling of interruptions. In this work, we designed a labeling sequence for our judge model to explicitly differentiate between floor-taking (successful interruption) and butting-in (unsuccessful interruption) interruptions. This enhancement ensures that our evaluation protocol comprehensively assesses all key aspects of human-AI turn-taking behavior.
> >
> > To conclude, we acknowledge the valuable contributions of prior work on training and evaluating turn-taking models. However, the prior evaluation metrics cannot be directly applied to assessing the quality of turn-taking decisions made by AI systems in human-AI conversations due to the absence of ground truth. Even when adapted using pseudo-ground truths or human relevance judgments, they may not be appropriate in certain scenarios or fully capture the complete range of capabilities needed to evaluate whether audio FMs can effectively manage turns in human-AI interactions. Based on your feedback, we will add this discussion to the paper.
> >
> > References
> >
> > [1] Ekstedt, Erik, and Gabriel Skantze. Voice activity projection: Self-supervised learning of turn-taking events.

---

> > > ### Author Response · Authors · 2024-11-25
> > > **Official comment by authors**
> > >
> > > Thanks Reviewer mo78 for your thorough review! We hope that information in our response helps clarify some of your concerns. We hope that you will take a look and consider updating your score.

---

> > > ### Comment · Reviewer_mo78 · 2024-12-02
> > >
> > > Thank you for your detailed response! My concerns have been partially addressed, and I will adjust the contribution score accordingly. However, I will maintain my overall rating due to ongoing concerns about the relatively low agreement between the majority of judge labels and human judgments, as highlighted by another reviewer.

---

> > > > ### Author Response · Authors · 2024-12-02
> > > >
> > > > Dear Reviewer mo78,
> > > >
> > > > Thank you for taking the time to provide thoughtful and constructive feedback. We sincerely appreciate your efforts and have tried to address your concerns regarding the agreement between judge labels and human decisions in our general response. We would like to emphasize that prior works have reported similar levels of agreement when using LLMs as judges, with around 70% [1] and 66% [2] agreements. We hope the clarifications we provided align with your expectations and address the issues raised comprehensively.
> > > >
> > > > As the discussion period deadline approaches, we kindly ask if you could take a moment to review our response. If you have any additional questions or require further elaboration, we would be grateful for the opportunity to address them promptly.
> > > >
> > > > Thank you once again for your valuable insights and guidance throughout this process. We deeply appreciate your time and support.
> > > >
> > > > References
> > > >
> > > > [1] Yiang et al. AIR-Bench: Benchmarking Large Audio-Language Models via Generative Comprehension (https://arxiv.org/pdf/2402.07729 )
> > > >
> > > > [2] Kolchinski et al. Approximating Human Judgment of Generated Image Quality. (https://arxiv.org/pdf/1912.12121 )

---

> > > > > ### Comment · Reviewer_mo78 · 2024-12-03
> > > > >
> > > > > Thank you for your response. I have considered the response. While some concerns have remained still (1) the evaluation protocol is highly expensive and (2) biased participants, most concerns are addressed and I agree that the evaluation protocol is promising.
> > > > >
> > > > > Therefore, I would raise the score.
> > > > >
> > > > > Additionally, I suggest including a detailed description of the training budgets and experimental setup in the main paper, as the judge model plays a crucial role in your protocol. This would further strengthen your work.
> > > > >
> > > > > Good luck with your submission!

---

> > > > > > ### Author Response · Authors · 2024-12-03
> > > > > >
> > > > > > Thank you for thoughtfully considering our response and for your constructive feedback. We are grateful for your positive recommendation and the updated score.
> > > > > >
> > > > > > We fully agree that providing a detailed description of the training budgets and experimental setup of the judge model will enhance the clarity of our work. We will ensure this is thoroughly addressed in the main paper!

---

### Official Review · Reviewer_BKtK · 2024-11-03

**Soundness:** 2
**Presentation:** 3
**Contribution:** 4
**Rating:** 6
**Confidence:** 4

**Summary:**

The paper presents an evaluation protocol designed to assess the turn-taking capabilities of spoken dialogue systems. It evaluates the exact timing of these events using a supervised model trained to predict them. The experimental results reveal interesting insights about existing spoken dialogue systems and offer valuable suggestions for their future development.

**Strengths:**

1. This paper proposes a comprehensive evaluation protocol and well-designed metrics to assess the turn-taking capabilities of spoken dialogue systems. The evaluation framework and metrics are thoughtfully developed and provide valuable insights.
2. The paper extends the evaluation of turn-taking capabilities of spoken dialogue systems from corpus-level statistics to a more granular assessment of the timing of turn-taking events. This fine-grained approach enables a more accurate reflection of a spoken dialogue system’s turn-taking capabilities.
3. The proposed evaluation metrics provide insights into the limitations of current systems in achieving interactive and natural conversations, highlighting areas for potential improvement.

**Weaknesses:**

1. In Metric (E), the judge labels show low consistency with human relevance judgments, indicating that this metric may have limited reliability in assessing the model's ability to handle user interruptions effectively.
2. My primary concern is the relatively low agreement between the majority of judge labels and human judgments, with most falling below 80%. This raises questions about the strength of the claim that the proposed metrics maintain high consistency with human decisions.
3. GPT-4o was not evaluated.

If my above concerns are resolved, I would consider increasing my rating.

**Questions:**

My questions are listed above.

---

> ### Author Response · Authors · 2024-11-21
> **Response to Reviewer BKtK**
>
> Thanks for the valuable comments and suggestions. We address your concerns below.
>
> ---
>
> # 1. GPT-4o was not evaluated.
>
> Please check the general response on evaluating more audio FMs.
>
> ---
>
> # 2. Agreement between the judge labels and human judgments
> > My primary concern is the relatively low agreement between the majority of judge labels and human judgments, with most falling below 80%. This raises questions about the strength of the claim that the proposed metrics maintain high consistency with human decisions.
>
> We acknowledge the reviewer’s concerns. We would like to emphasize that turn-taking prediction is indeed a challenging task, as turn-taking behavior can vary widely even among different users. Despite this complexity, our turn-taking model achieves performance comparable to those reported in prior studies, which we believe demonstrates its robustness and practical utility. Our model also maintains moderate alignment with human decisions even on OOD spoken conversation datasets—an indicator of its generalizability.
>
> That said, we would like to stress that the primary contribution of this work lies in our novel evaluation protocol, which is designed to be adaptable and can integrate any turn-taking prediction model. We agree that enhancing the model’s accuracy would further improve the protocol's reliability, and explicitly discuss this in Sec. 6 (Limitations) in the main text.
>
> ---
>
> # 3. Low consistency with human  judgments for Metric (E)
> > In Metric (E), the judge labels show low consistency with human relevance judgments, indicating that this metric may have limited reliability in assessing the model's ability to handle user interruptions effectively.
>
> This is indeed a limitation of our work and we have acknowledged this explicitly in section 4.8.  Moving forward, we consider this an area for further improvement, where enhancing the judge model could lead to better alignment with human judgments.

---

> > ### Author Response · Authors · 2024-11-25
> > **Official Comment**
> >
> > Thanks Reviewer BKtK for your thorough review! We hope that information in our response helps clarify some of your concerns. We hope that you will take a look and consider updating your score.

---

> > ### Comment · Reviewer_BKtK · 2024-11-26
> > **Response to Authors**
> >
> > Thanks for your response. The additional experiments on GPT-4o reveal interesting insights about the model, and the direction of this work is promising. However, since my primary concern has not been addressed, I'll keep my current rating.

---

> > > ### Author Response · Authors · 2024-11-26
> > >
> > > Thank you for considering our response!

---

> > > > ### Author Response · Authors · 2024-12-02
> > > >
> > > > Dear Reviewer BKtK,
> > > >
> > > > Thank you for taking the time to provide thoughtful and constructive feedback. We sincerely appreciate your efforts and have tried to address your concerns regarding the agreement between judge labels and human decisions in our general response. We hope the clarifications we provided align with your expectations and address the issues raised comprehensively.
> > > >
> > > > As the discussion period deadline approaches, we kindly ask if you could take a moment to review our response. If you have any additional questions or require further elaboration, we would be grateful for the opportunity to address them promptly.
> > > >
> > > > Thank you once again for your valuable insights and guidance throughout this process. We deeply appreciate your time and support.

---

### Official Review · Reviewer_cJr2 · 2024-11-04

**Soundness:** 2
**Presentation:** 2
**Contribution:** 2
**Rating:** 3
**Confidence:** 3

**Summary:**

This paper proposes an evaluation protocol to measure the turn-taking capabilities of audio foundation models.

**Strengths:**

The strengths of this paper are as follows:
 1. This paper provides an automated turn-taking protocol for audio foundation models
 2. The evaluation platform will be open-sourced.

**Weaknesses:**

The weaknesses of this paper are as follows:
 1. The study aims to measure precise turn-taking, but the thresholds are set to arbitrary values.
 2. The participants introduced in Sec 3 seem biased, consisting of the authors and related individuals.
 3. The confidence in some evaluations (Fig. 3(b), (e)) appears high, but no explanation is provided.

**Questions:**

Here are questions for the authors:
 - The thresholds in Sec. 4.4-4.8 seem arbitrary. Is there a specific reason for choosing these values? All units appear to represent likelihoods, yet they range from negative ($threshold_3$ = -0.45) to positive values ($threshold_2$ = 0.1).
 - There are concerns about the reliability of the judge model. Since all results are based on comparisons with this model, is there concrete evidence supporting its credibility? Specifically, the conclusion that Moshi[1] is "too aggressive" lacks persuasiveness if it relies solely on comparisons with the judge model.

[1] Defossez et al. Moshi: a speech-text foundation model for real-time dialogue

---

> ### Author Response · Authors · 2024-11-21
> **Response to Reviewer cJr2**
>
> Thank you for your insightful comments. We address your concerns below.
>
> ---
>
> # 1. Thresholds for proposed metrics
> > The thresholds in Sec. 4.4-4.8 seem arbitrary. Is there a specific reason for choosing these values?
>
> We apologize if this was not sufficiently clear. As mentioned in Sec 4.3 (lines 307-309) on page 6, we carefully tuned the thresholds for all proposed metrics using an in-domain validation set to maximize the agreement between the judge model's labels and human judgments. We present the agreement between judge labels and human judgments on the in-domain validation set in Table 6 in the Appendix. We hope this addresses your concern.
>
> ---
>
> # 2. Reliability of judge model
> >There are concerns about the reliability of the judge model. Since all results are based on comparisons with this model, is there concrete evidence supporting its credibility?
>
> We agree that establishing the reliability of the judge model is essential, given that our results depend on using the judge label. To address this, we outline our validation approach in Sec. 4.3 (lines 302-312) on page 6, inspired by prior works [1, 2] showing high consistency between LLM predictions and human relevance judgments.
>
> In our study, we assess judge model consistency with human judgments by analyzing instances in a human-human conversation dataset corresponding to each metric. For example, Metric A considers scenarios where a listener decides whether to speak up during a speaker’s pause. Detailed explanations for all metrics are in sections 4.4-4.8 (e.g., lines 322-323 for Metric A, lines 374-375 for Metric B, lines 399-401 for Metric C, lines 414-415 for Metric D, and lines 448-450 for Metric E). As shown in Figure 3 on page 7, our judge labels have good agreement with human judgments on both in-domain (blue) and out-of-domain (green) test sets for most metrics, supporting the model's credibility. We hope this clarifies our approach.
>
> ---
>
> # 3. Big Confidence Intervals for Proposed Metrics
> >The confidence in some evaluations (Fig. 3(b), (e)) appears high, but no explanation is provided.
>
> Thank you for pointing out this important observation. We acknowledge the reviewer’s concern regarding the high confidence intervals for Metric (B) in Fig. 3(b) and Metric (E) in Fig. 3(e). We hypothesize that this result from a small sample size, as shown in Table 2, where occurrences of the AI system backchanneling (0.01% for both Moshi and Cascaded systems) and user interruptions (0.2% for Moshi and 0.1% for the Cascaded system) are indeed very rare in human-AI conversations. We updated the paper (lines 387-389 in Sec. 4.5 and lines 458-460 in Sec. 4.8) to clearly address and discuss this.
>
> ---
>
> # 4. Biased participants
> >The participants introduced in Sec 3 seem biased, consisting of the authors and related individuals.
>
> Thank you for raising this important concern. We acknowledge the potential for bias in our user study, as participants were primarily the authors and their research colleagues. Similar “lab experiments” are common in prior work [3], enabling focused testing of specific functionalities (eg. prompting participants to interrupt the AI system) and facilitating deeper insights through discussions about their subjective experiences. However, we agree that this controlled setup may introduce bias. We hope our evaluation protocol inspires future studies to conduct larger-scale evaluations with more diverse participants. We explicitly discuss and acknowledge this limitation in Sec. 6 and A.2 in the paper.
>
> ---
>
> References
>
> [1] AIR-Bench: Benchmarking Large Audio-Language Models via Generative Comprehension (https://arxiv.org/abs/2402.07729)
>
> [2] Judging LLM-as-a-Judge with MT-Bench and Chatbot Arena (https://arxiv.org/pdf/2306.05685)
>
> [3] Survey on evaluation methods for dialogue systems [https://link.springer.com/article/10.1007/s10462-020-09866-x]

---

> ### Comment · Reviewer_cJr2 · 2024-11-25
>
> ### Threshold and Reliability
>
> **L303-304**
> > we take inspiration from prior works (Yang et al., 2024; Zheng et al.,2024) that have experimented with using an LLM as a judge.
>
> When considering the concept of an LLM as a judge, it is customary to employ a robust model such as GPT-4. Typically, these models demonstrate agreement rates exceeding 80% when compared with human evaluations [1]. Furthermore, such models are validated not only against subjective measures like agreement but also through objective benchmarks [2]. These high standards highlight the importance of evaluating how well the proposed approach aligns with them.
>
> Although the proposed model shows high agreement with human judgments within the 'dataset' used for human-human comparisons, this does not necessarily establish the model as 'objectively' strong. Compared to the numerous benchmarks used to evaluate LLMs, the evidence provided in Sections 4.4–4.8 (e.g., lines 322–323 for Metric A, lines 374–375 for Metric B, lines 399–401 for Metric C, lines 414–415 for Metric D, and lines 448–450 for Metric E) may not be sufficient to conclude that it is an excellent judge. That said, I want to stress that this critique does not imply the evaluation methods themselves are flawed.
>
> > we carefully tuned the thresholds ... to maximize the agreement between the judge model's labels and human judgments.
>
> Therefore, a good judge model should minimize the need for extensive hyperparameter tuning to achieve consistent results. Alternatively, it might be beneficial to first demonstrate that the model's performance is robust and consistent across different hyperparameter settings.
>
> [1] Zheng et al., Judging LLM-as-a-Judge with MT-Bench and Chatbot Arena https://arxiv.org/abs/2306.05685
>
> [2] OpenAI, https://arxiv.org/abs/2303.08774
>
> Finally, I have adjusted the contribution score to reflect the strengths of this work, particularly its thoughtful exploration of evaluation techniques and efforts to align human and model judgments. While there are areas for improvement, the study provides valuable insights that advance discussions in this domain.

---

> > ### Author Response · Authors · 2024-11-25
> > **Official Comment by Authors**
> >
> > > Furthermore, such models are validated not only against subjective measures like agreement but also through objective benchmarks
> >
> > We would like to emphasize that we also evaluate using objective metrics as shown in Table 1. Our results clearly demonstrate that our model performs **on par with prior turn-taking prediction models**. Additionally, our model exhibits strong out-of-domain (OOD) generalization. Specifically, it achieves robust **zero-shot performance on two OOD** datasets: (1) non-task-oriented spoken dialogues (Fisher Corpus) and (2) task-oriented spoken dialogues (Columbia Games Corpus).
> >
> > That said, we would like to stress that the primary contribution of this work lies in our novel evaluation protocol, which is designed to be adaptable and can **integrate any turn-taking prediction model**. Turn-taking prediction is indeed a challenging task and enhancing the model’s accuracy would further improve the protocol's reliability. We explicitly discuss this in Sec. 6 (Limitations) in the main text.
> >
> > ----
> >
> > > Therefore, a good judge model should minimize the need for extensive hyperparameter tuning to achieve consistent results. Alternatively, it might be beneficial to first demonstrate that the model's performance is robust and consistent.
> >
> > It is important to understand that **number of turn taking events are extremely unbalanced** with **continuation (C)** and **silence (NA)** making up the label set for more than 95% of instances. As a result, it is indeed a **common practice [1] in prior literature** to tune the threshold for each label on the validation set. However, our results show that these threshold not only lead to consistently good agreement with human judgement on an in-domain test but also **generalise well to an out-of-domain test** set showing that our model is robust and achieves consistent results even in out of domain setting.
> >
> > ---
> >
> > We hope that information in our response helps clarify some of your concerns on the robustness and reliability of our approach. We hope that you will take a look and consider updating your score.
> >
> > References
> >
> > [1] Ekstedt, Erik, and Gabriel Skantze. Voice activity projection: Self-supervised learning of turn-taking events.

---

> > > ### Author Response · Authors · 2024-11-26
> > > **Update**
> > >
> > > **Requirement for threshold tuning for judge model**
> > > > it might be beneficial to first demonstrate that the model's performance is robust and consistent across different hyperparameter settings.
> > >
> > > In response to your comment, we surveyed prior literature to better quantify the impact of threshold tuning on our turn-taking model’s performance and identified two main approaches:
> > >
> > > 1. Sensitivity Analysis [1,2,3]: Prior works have experimented with varying key hyperparameters systematically over a range and analyzed validation performance as a function of hyperparameters.
> > > Inspired by this, we vary thresholds from -0.5 to 0.5 for metrics where judge labels are computed using the difference of 2 likelihoods (all metrics except metric B in Eq. 3) and from 0 to 1 for others (i.e., for metric B). The thresholds are incremented in steps of 0.01. We then calculated the Margin of Error (ME) [4] for 95% Confidence Intervals to quantify uncertainty in agreement between judge labels and human relevance judgments due to sensitivity to threshold values.
> > > The margin of error (ME) is calculated as:
> > >
> > > $ME = z \cdot \frac{\sigma}{\sqrt{n}}$
> > >
> > > Where:
> > > - \(z = 1.96\): Z-score for a 95% confidence level.
> > > - \($\sigma$\): Standard deviation
> > > - \(n\): Sample size (i.e. Size of threshold range over which performance is computed =100 for our experiments)
> > >
> > > The observed margin of errors, along with the agreement of the judge label with human decisions on the in-domain validation set, are shown below.
> > >
> > > | **Turn-Taking Metric**                           | **Agreement with Judge Label** | **Margin of Error (ME)** |
> > > |--------------------------------------------------|------------------------------|-------------------|
> > > | **Metric A and D: When to speak up?**           |                                 |                     |
> > > | When listener decides to speak up               | 81.2                           | ±6.43                |
> > > | When listener lets speaker continue             | 75.5                           | ±7.46                |
> > > | **Metric B: When to backchannel?**              |                                 |                     |
> > > | When listener backchannels                      | 71.6                           | ±5.75                |
> > > | When listener does not backchannel              | 74.8                           | ±3.49                |
> > > | **Metric C: When to interrupt?**                |                                 |                     |
> > > | When listener interrupts                        | 76.6                           | ±5.13                |
> > > | When listener does not interrupt                | 72.8                           | ±1.80                |
> > > | **Metric E: Handle user interruptions?**        |                                 |                     |
> > > | When interrupting speaker takes the floor       | 61.3                           | ±5.90                |
> > > | When interrupting speaker does not take the floor| 57.0                          | ±5.30                |
> > >
> > > Generally, a margin of error of below 5% is considered excellent for high-accuracy needs, and less than 10% is acceptable for most studies. Our analysis shows that the agreement with human judgements does not undergo huge fluctuations with changes in threshold (i.e., margin of error is always less than 10%), and hence, our approach does not require extensive hyperparameter tuning. Based on your feedback, we will add these results to Table 6 in the paper.
> > >
> > > 2. Validation on Multiple Datasets [5]: Prior works have argued the reliability of their model by showing consistent performance across datasets, demonstrating that the model generalizes well without dataset-specific tuning. In Figure 3, our judge labels achieve good agreement with human judgment even on the OOD spoken dialog dataset, i.e., Columbia Games Corpus, without a threshold being specifically tuned for this dataset. This result shows that our model achieves consistent performance without dataset-specific tuning.
> > >
> > > Our new analysis provides evidence of the robustness and consistency of our model. We hope that these findings address your concerns regarding the reliability of our model, and we will incorporate these results into the paper.
> > >
> > > ---
> > >
> > > References
> > >
> > > [1] Novello et al. Goal-Oriented Sensitivity Analysis of Hyperparameters in Deep Learning.
> > >
> > > [2] Razavi et al. The Future of Sensitivity Analysis: An essential discipline for systems modeling and policy support.
> > >
> > > [3] Sadeghi et al. A Review of Global Sensitivity Analysis Methods and a comparative case study on Digit Classification
> > >
> > > [4] Tanur et al. Margin of error.
> > >
> > > [5] Rijn et al. Hyperparameter Importance Across Datasets.

---

> > > > ### Author Response · Authors · 2024-12-02
> > > >
> > > > Dear Reviewer cJr2,
> > > >
> > > > Thank you for taking the time to provide thoughtful and constructive feedback. We sincerely appreciate your efforts and have tried to address your concerns in our general response as well as previous responses. We hope the clarifications we provided align with your expectations and address the issues raised comprehensively.
> > > >
> > > > As the discussion period deadline approaches, we kindly ask if you could take a moment to review our response. If you have any additional questions or require further elaboration, we would be grateful for the opportunity to address them promptly.
> > > >
> > > > Thank you once again for your valuable insights and guidance throughout this process. We deeply appreciate your time and support.

---

### Official Review · Reviewer_Ddi6 · 2024-11-04

**Soundness:** 3
**Presentation:** 3
**Contribution:** 3
**Rating:** 6
**Confidence:** 3

**Summary:**

This paper addresses the challenges of evaluating the turn-taking capabilities of audio foundation models (FMs) in conversational settings.
It defines 6 types of Turn-Taking Events and evaluates the performance of end-to-end speech dialogue models as well as cascaded systems.
Through the results obtained from this study, the authors discovered numerous issues with existing AI dialogue systems in handling turn-taking, such as sometimes failing to intervene in conversations at appropriate times or excessively interrupting others. Furthermore, the authors conducted tests on multiple open-source and closed-source audio foundation models, revealing their limitations in understanding and predicting turn-taking events, and highlighting areas for improvement.

**Strengths:**

1. The definition of turn-taking is detailed and clear.

2. The evaluation protocol proposed in this paper contributes to better assessing the performance of audio foundation models in dialogues, providing strong support for the development of voice dialogue systems.

3. This paper reveals many issues existing in current AI dialogue systems when handling turn-taking, offering valuable references for future research.

**Weaknesses:**

1. This study only tested a few open-source and closed-source audio FM models.

2. There is a lack of comprehensive performance evaluation and summary.

**Questions:**

NA.

---

> ### Author Response · Authors · 2024-11-21
> **Response to Reviewer Ddi6**
>
> Thanks for the insightful suggestions and acknowledging that our work offers valuable references for future development of voice dialogue systems. We address your concerns below.
>
> ---
>
> # 1. Evaluated few audio FMs
> > This study only tested a few open-source and closed-source audio FM models.
>
> Check general response
>
> ---
>
> # 2. There is a lack of comprehensive performance evaluation and summary.
>
> We apologize if our comprehensive evaluation was not clearly emphasized in the draft. Our work provides a thorough assessment of spoken dialogue systems by reviewing prior literature to identify key turn-taking abilities for effective human-AI interaction and developing specific metrics (Sections 4.4-4.8) to evaluate these skills. Our analysis highlights significant limitations in current AI dialogue systems and outlines key research directions for improvement, as noted by reviewers (@R Ddi6, @R BKtK, @R mo78). If you have specific suggestions for additional evaluations, we will try to incorporate them into the paper.
>
> ---

---

> > ### Author Response · Authors · 2024-11-25
> > **Official Comment**
> >
> > Thanks Reviewer Ddi6 for your thorough review! We hope that information in our response helps clarify some of your concerns. We hope that you will take a look and consider updating your score.

---

### Official Review · Reviewer_WGPU · 2024-11-04

**Soundness:** 3
**Presentation:** 3
**Contribution:** 3
**Rating:** 8
**Confidence:** 3

**Summary:**

The paper proposes a new evaluation protocol to assess the spoken dialog system's turn-taking capabilities, i.e., the Moshi and Cascaded model. They use a supervised model as a judge which is trained to predict turn-tasking events in human-human conversation (i.e., Switchboard). The paper presents a comprehensive user study that evaluates the Moshi and Casaded model on their ability to perform turn-taking events, and it finds that they sometimes do not understand when to speak up, can interrupt too aggressively, and rarely backchannel. The main contributions are:
1. A new evaluation protocol to assess the spoken dialog system's turn-taking capabilities.
2. Some insight about existing spoken dialogue systems through user study.
3. Additionally create a test benchmark using Switchboard dataset to evaluate SALMONN, Qwen2-audio-instruct, Qwen-audiochat, Whisper+GPT-4o on their ability to understand and predict turn-taking events.

**Strengths:**

1. The originality of the work is commendable. The authors propose a novel evaluation protocol to assess the turn-taking capabilities of spoken dialog systems.
2. The paper is well-written and provides sufficient experimental details in the Appendix.
3. The authors plan to open source the evaluation platform.

**Weaknesses:**

1. The evaluation protocol is highly expensive, as it requires a supervised dataset to train the judge model. This approach is not feasible if we lack a supervised dataset in other languages, such as Chinese.
2. The filler word set for backchannel detection is heuristic and may miss some backchannel cases that are not included in the filler word set.

**Questions:**

1. The Fisher dataset is a common dataset comparable to Switchboard. What is the performance of the supervised turn-taking prediction model on this dataset?
2. How can your evaluation protocol be adapted or applied in scenarios where supervised datasets are not available for other languages, such as Chinese?

---

> ### Author Response · Authors · 2024-11-21
> **Response to Reviewer WGPU**
>
> Thank you for your valuable comments and acknowledging the originality of our work. Weakness 3 and 4 are similar to weakness 1 and 2 and we address remaining concerns below.
>
> ---
>
> # 1. Results on Fisher:
>
> Please check general response.
>
> ---
>
> # 2. Application to other languages:
> >“How can your evaluation protocol be adapted or applied in scenarios where supervised datasets are not available for other languages, such as Chinese?”
>
> Please check general response on scalability and applicability of our approach.
>
> ---
>
> # 3. Backchannel detection is heuristic
> > The filler word set for backchannel detection is heuristic. It may ignore some backchannel case not in filler word set.
>
> Thank you for your insightful observation. We acknowledge that our approach for identifying backchannels relies on heuristics, using common one- and two-word phrases as indicators of backchannels. While this may miss some backchannels, it aligns with standard practices in prior turn-taking models [1, 2], which also use similar heuristic methods. Based on your comment, we explicitly discussed this limitation in Sec. 6 in the main paper.
>
> ---
>
> References
>
> [1] TurnGPT: a Transformer-based Language Model for Predicting Turn-taking in Spoken Dialog (https://aclanthology.org/2020.findings-emnlp.268/)
>
> [2] Turn-taking and Backchannel Prediction with Acoustic and Large Language Model Fusion (https://arxiv.org/abs/2401.14717)

---

> > ### Author Response · Authors · 2024-11-25
> > **Official Comment**
> >
> > Thanks Reviewer WGPU for your thorough review! We hope that information in our response helps clarify some of your concerns. We hope that you will take a look and consider updating your score.

---

### Author Response · Authors · 2024-11-21
**General Response**

We thank all the reviewers for their thoughtful feedback. We are pleased that our approach is recognized as novel (@R WGPU, @R mo78), thoughtfully designed (@R BKtK), and clearly presented (@R WGPU, @R Ddi6). We appreciate the recognition of its value in assessing conversational capabilities of audio FMs (@R Ddi6, @R BKtK), highlighting AI dialogue system limitations, and suggesting future research directions (@R Ddi6, @R BKtK, @R mo78). We will incorporate the reviewers’ constructive suggestions into the paper.

---

# 1. Results on Fisher

Reviewers (@R WGPU, @R mo78) suggested evaluating our judge turn-taking model on an out-of-domain, non-task-oriented spoken dialogue dataset, such as Fisher. To address this, we created a random test set comprising 23 hours of audio and 138 conversations, similar in size to Switchboard test set, and we will make this test split publicly available. The Fisher dataset's transcriptions were created using the Quick Transcription specification, which introduced inaccuracies and left significant portions untranscribed as also noted in [1]. We developed heuristics to identify such audio segments with large untranscribed content and made sure to exclude these audio segments from the test set. Further manual analysis revealed errors in the ground truth timestamps and we corrected these timestamps using speaker diarization outputs from Pyannote.

Our results show that our model demonstrates strong generalization on this corpus, achieving a **ROC-AUC of 91.0** comparable to the performance on the in-domain test set, i.e., Switchboard (overall ROC-AUC score of 92.0). These results and a detailed discussion have been included in the updated draft (Tab. 1, Sec. 4.1, Sec A.4), and we will make all our data preparation code and model publicly available. Thank you for the suggestion, as it helped us further validate the robustness of our approach.

---

# 2. Evaluating more audio FMs
We acknowledge the reviewer's (@R Ddi6, @R BKtK) concern about benchmarking only a limited set of audio FMs for evaluating conversational capabilities. At the time of conducting this study, there were indeed very few audio FMs capable of performing turn-taking events, limiting our selection. We emphasize that this is a developing field, and we plan to expand our benchmarking as more audio FMs with turn-taking capabilities emerge. We explicitly mention this in the updated paper (Limitations in Sec. 6).

@R BKtK mentioned that we should have evaluated GPT-4o. At the time of submission, GPT-4o’s Advanced Voice Mode was not publicly available, preventing its inclusion in our evaluations. Based on the feedback, we have started collecting human-GPT-4o conversation data, accumulating 55 minutes and 38 seconds of audio across six speakers. This preliminary test trial has revealed some interesting insights:

- Turn-Taking Latency: GPT-4o exhibits a moderate gap between speaker turns (16.1% of cumulative duration), smaller than the Cascaded system (32.5% in Fig. 2b) but larger than Moshi (11.8% in Fig. 2b), indicating intermediate latency.
- Overlap: Similar to a cascaded system, GPT-4o has minimal overlapping speech (0.5% of cumulative duration in Fig. 2b), resulting in less interactive conversations.
- Turn-Yielding Behavior: GPT-4o has a high number of pause events (18.0 per minute) and fewer gap events (3.2 per minute) compared to the other dialogue systems (Fig. 2a), indicating that it sometimes speaks for very long without yielding it’s turn, which makes the conversation bland and less engaging for end users.
- Metric C (Sec. 4.6): GPT-4o rarely interrupts users (0.1%, lower than the other systems in Table 2), but when it does, its agreement with the judge label (75.0%) is significantly higher than Moshi (35.7% in Fig. 3c) and the Cascaded system (24.2% in Fig. 3c).
- Metric D (Sec. 4.7): GPT-4o is significantly better at conveying users when it wants to keep the conversation floor, having 68.9% agreement with judge label which is much higher than Moshi (32.7% agreement in Fig. 3d) and Cascaded system (40.8% agreement in Fig. 3d).

We are currently expanding this effort to collect 4 hours of GPT-4o conversation data, comparable to the results reported for other dialogue systems in the draft. This analysis with GPT-4o will be included in the final paper.

We agree with the reviewers (@R Ddi6, @R BKtK, @R mo78) that, while our analysis is limited to a few audio FMs, it effectively highlights issues in current AI dialogue systems, offering valuable insights for future research and underscoring the utility of our protocol.

---

References:

[1] Generative spoken dialogue language modeling (https://doi.org/10.1162/tacl_a_00545 )

---

> ### Author Response · Authors · 2024-11-21
> **General Response**
>
> # 3. Scalability and Applicability of Approach
>
> Reviewers (@R WGPU, @R mo78) mentioned that the evaluation protocol needs a supervised dataset to train the judge model, limiting its scalability and applicability. We agree that this is a limitation of our approach and updated the draft to explicitly acknowledge it.  While our current approach indeed relies on a supervised dataset to train a turn-taking model as the judge, this model can be trained on any spoken conversation dataset containing speaker turns, transcripts, and timestamps, which are often available even for non-English languages. As noted in Appendix A.1, prior work [2] successfully trained turn-taking models on Chinese (Mandarin) and Japanese using publicly available datasets [3, 4]. Interestingly, multilingual models trained on English, Chinese, and Japanese perform comparably to monolingual models despite the diverse turn-taking behaviors of these languages.
>
> To address scenarios without supervised datasets, we propose a low-cost solution: collecting a small spoken dataset for the target language and generating annotations through human efforts or using tools like PyAnnote (speaker diarization) and Whisper (ASR). We can then train multilingual turn-taking models that leverage high-resource language data to improve performance on low-resource languages. This approach is far more cost-effective than collecting human relevance judgments for every turn-taking event. In response to your feedback, we added a discussion in the paper (Limitations in Sec. 6, A.7) on adapting our evaluation to non-English and low-resource languages.
>
> ---
>
> References:
>
> [2] Multilingual Turn-taking Prediction Using Voice Activity Projection (https://aclanthology.org/2024.lrec-main.1036.pdf )
>
> [3] HKUST Mandarin Telephone Speech (https://catalog.ldc.upenn.edu/LDC2005S15 )
>
> [4] Japanese Travel Agency Task Dialogues (https://aclanthology.org/2022.lrec-1.619/)

---

### Author Response · Authors · 2024-12-02
**General Response regarding the agreement between judge labels and human decisions**

We sincerely thank the reviewers for their thoughtful feedback. Some reviewers (@R cJr2, @R BKtK, @R mo78). raised concerns about the relatively low agreement (<80%) between judge labels and human judgments. We would like to clarify that, with the exception of Metric E, the judge labels demonstrate a consistently high level of agreement (>70%, as shown in Figure 3) with human decisions on the in-domain test set. Additionally, prior works have reported **similar levels of agreement when using LLMs as judges, with around 70% [1] and 66% [2] agreements**. Turn-taking, by nature, is a subjective task influenced by individual user behavior, making complete alignment neither expected nor feasible. Moreover, our judge labels also show moderate agreement with human decisions even on an out-of-domain test set.

To further demonstrate the robustness of our trained turn-taking model, we evaluated it objectively on its ability to predict upcoming turn-taking events in human-human conversations. This evaluation was conducted on both an in-domain test set (Switchboard) and two out-of-domain datasets (Columbia Games and Fisher). Our results indicate that the model not only performs on par with prior works but also generalizes strongly to OOD spoken dialogue corpora in zero shot manner, achieving comparable performance across datasets.

These findings suggest that the model can reliably evaluate the precise timing of turn-taking decisions made by AI dialogue systems. Its decent consistency with human decisions indicates that it serves as an effective proxy for human judgment, addressing concerns regarding its reliability and utility. We will incorporate these discussions into the paper.

References

[1] Yiang et al. AIR-Bench: Benchmarking Large Audio-Language Models via Generative Comprehension (https://arxiv.org/pdf/2402.07729 )

[2] Kolchinski et al. Approximating Human Judgment of Generated Image Quality. (https://arxiv.org/pdf/1912.12121 )

---

### Meta-Review · Area_Chair_MGh6 · 2024-12-18

**Metareview:**

This paper proposes a novel evaluation protocol for assessing turn-taking capabilities in audio-based spoken dialogue systems and audio foundation models. By defining metrics to capture when to speak, backchannel, interrupt, and handle interruptions, the authors introduce a supervised judge model trained on human-human conversations. Reviewers appreciated the motivation and the effort to formalize a challenging, subjective aspect of conversational interaction. They also welcomed the clarification that while perfect human-model alignment is unrealistic, the judge model achieves moderate to high agreement with human judgments across datasets. Although some thresholds and methods could be justified, and certain baselines were missing, additional evaluations indicate that the approach is sound and generalizes to out-of-domain scenarios.
Given these points and the constructive improvements offered by the authors, I recommend acceptance.

**Additional Comments On Reviewer Discussion:**

During the discussion, the authors clarified concerns about threshold choices and demonstrated that their turn-taking judge model generalizes well to out-of-domain datasets. They compared their agreement rates with human judgments to existing literature, showing comparable performance.  The reviewers generally agree that this contribution offers a valuable step forward in evaluating conversational turn-taking, supporting the decision to accept.

---

### Decision · Program_Chairs · 2025-01-22

Accept (Poster)